# PATH INTEGRAL SAMPLER:
# A STOCHASTIC CONTROL APPROACH FOR SAMPLING

**Qinsheng Zhang**
Center for Machine Learning
Georgia Institute of Technology
qzhang419@gatech.edu

**Yongxin Chen**
School of Aerospace Engineering
Georgia Institute of Technology
yongchen@gatech.edu

## ABSTRACT

We present Path Integral Sampler (PIS), a novel algorithm to draw samples from unnormalized probability density functions. The PIS is built on the Schrödinger bridge problem which aims to recover the most likely evolution of a diffusion process given its initial distribution and terminal distribution. The PIS draws samples from the initial distribution and then propagates the samples through the Schrödinger bridge to reach the terminal distribution. Applying the Girsanov theorem, with a simple prior diffusion, we formulate the PIS as a stochastic optimal control problem whose running cost is the control energy and terminal cost is chosen according to the target distribution. By modeling the control as a neural network, we establish a sampling algorithm that can be trained end-to-end. We provide theoretical justification of the sampling quality of PIS in terms of Wasserstein distance when sub-optimal control is used. Moreover, the path integrals theory is used to compute importance weights of the samples to compensate for the bias induced by the sub-optimality of the controller and time-discretization. We experimentally demonstrate the advantages of PIS compared with other start-of-the-art sampling methods on a variety of tasks.

## 1 INTRODUCTION

We are interested in drawing samples from a target density $\hat{\mu} = Z\mu$ known up to a normalizing constant $Z$. Although it has been widely studied in machine learning and statistics, generating asymptotically unbiased samples from such unnormalized distribution can still be challenging (Talwar, 2019). In practice, variational inference (VI) and Monte Carlo (MC) methods are two popular frameworks for sampling.

Variational inference employs a density model $q$, from which samples are easy and efficient to draw, to approximate the target density (Rezende & Mohamed, 2015; Wu et al., 2020). Two important ingredients for variational inference sampling include a distance metric between $q$ and $\hat{\mu}$ to identify good $q$ and the importance weight to account for the mismatch between the two distributions. Thus, in variational inference, one needs to access the explicit density of $q$, which restricts the possible parameterization of $q$. Indeed, explicit density models that provide samples and probability density such as Autoregressive models and normalizing flow are widely used in density estimation (Gao et al., 2020a; Nicoli et al., 2020). However, such models impose special structural constraints on the representation of $q$. For instance, the expressive power of normalizing flows (Rezende & Mohamed, 2015) is constrained by the requirements that the induced map has to be bijective and its Jacobian needs to be easy-to-compute (Cornish et al., 2020; Grathwohl et al., 2018; Zhang & Chen, 2021).

Most MC methods generate samples by iteratively simulating a well-designed Markov chain (MCMC) or sampling ancestrally (MacKay, 2003). Among them, Sequential Monte Carlo and its variants augmented with annealing trick are regarded as state-of-the-art in certain sampling tasks (Del Moral et al., 2006). Despite its popularity, MCMC methods may suffer from long mixing time. The short-run performance of MCMC can be difficult to analyze and samples often get stuck in local minima (Nijkamp et al., 2019; Gao et al., 2020b). There are some recent works exploring the possibility of incorporating neural networks to improve MCMC (Spanbauer et al., 2020; Li et al., 2020b). However, evaluating existing MCMC empirically, not to say designing an objective loss

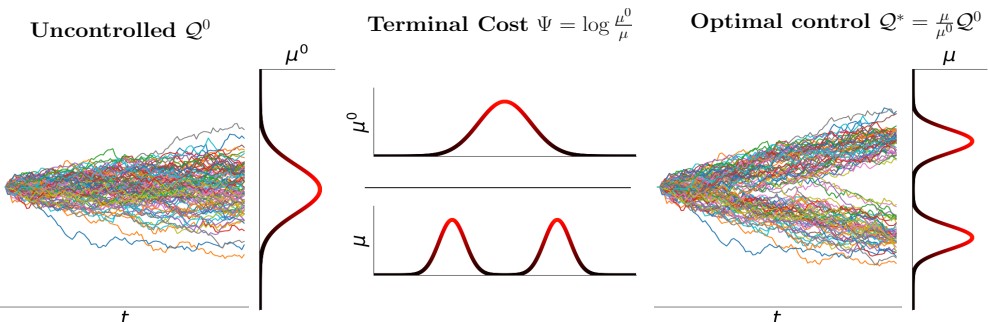

Figure 1: Illustration of Path Integral Sampler (PIS). The optimal policy of a specific stochastic control problem where a terminal cost function is chosen according to the given target density $\mu$, can generate unbiased samples over a finite time horizon.

function to train network-powered MCMC, is difficult (Liu et al., 2016; Gorham & Mackey, 2017). Most existing works in this direction focus only on designing data-aware proposals (Song et al., 2017; Titsias & Dellaportas, 2019) and training such networks can be challenging without expertise knowledge in sampling.

In this work, we propose an efficient sampler termed Path Integral Sampler (PIS) to generate samples by simulating a stochastic differential equation (SDE) in finite steps. Our algorithm is built on the Schrödinger bridge problem (Pavon, 1989; Dai Pra, 1991; Léonard, 2014; Chen et al., 2021) whose original goal was to infer the most likely evolution of a diffusion given its marginal distributions at two time points. With a proper prior diffusion model, this Schrödinger bridge framework can be adopted for the sampling task. Moreover, it can be reformulated as a stochastic control problem (Chen et al., 2016) whose terminal cost depends on the target density $\hat{\mu}$ so that the diffusion under optimal control has terminal distribution $\hat{\mu}$. We model the control policy with a network and develop a method to train it gradually and efficiently. The discrepancy of the learned policy from the optimal policy also provides an evaluation metric for sampling performance. Furthermore, PIS can be made unbiased even with sub-optimal control policy via the path integral theorem to compute the importance weights of samples. Compared with VI that uses explicit density models, PIS uses an implicit model and has the advantage of free-form network design. The explicit density models have weaker expressive power and flexibility compared with implicit models, both theoretically and empirically (Cornish et al., 2020; Chen et al., 2019; Kingma & Welling, 2013; Mohamed & Lakshminarayanan, 2016). Compared with MCMC, PIS is more efficient and is able to generate high-quality samples with fewer steps. Besides, the behavior of MCMC over finite steps can be analyzed and quantified. We provide explicit sampling quality guarantee in terms of Wasserstein distance to the target density for any given sub-optimal policy.

Our algorithm is based on Tzen & Raginsky (2019), where the authors establish the connections between generative models with latent diffusion and stochastic control and justify the expressiveness of such models theoretically. How to realize this model with networks and how the method performs on real datasets are unclear in Tzen & Raginsky (2019). Another closely related work is Wu et al. (2020); Arbel et al. (2021), which extends Sequential Monte Carlo (SMC) by combining deterministic normalizing flow blocks with stochastic MCMC blocks. To be able to evaluate the importance weights efficiently, MCMC blocks need to be chosen based on annealed target distributions carefully. In contrast, in PIS one can design expressive architecture freely and train the model end-to-end without the burden of tuning MCMC kernels, resampling or annealing scheduling. We summarize our contributions as follows. 1). We propose Path Integral Sampler, a generic sampler that generates samples through simulating a target-dependent SDE which can be trained with free-form architecture network design. We derive performance guarantee in terms of the Wasserstein distance to the target density based on the optimality of the learned SDE. 2). An evaluation metric is provided to quantify the performance of learned PIS. By minimizing such evaluation metric, PIS can be trained end-to-end. This metric also provides an estimation of the normalization constants of target distributions. 3). PIS can generate samples without bias even with sub-optimal SDEs by assigning importance weights using path integral theory. 4). Empirically, PIS achieves the state-of-the-art sampling performance in several sampling tasks.

## 2 SAMPLING AND STOCHASTIC CONTROL PROBLEMS

We begin with a brief introduction to the sampling problem and the stochastic control problem. Throughout, we denote by $\tau = \{\mathbf{x}_t, 0 \leq t \leq T\}$ a continuous-time stochastic trajectory.

### 2.1 SAMPLING PROBLEMS

We are interested in drawing samples from a target distribution $\mu(\mathbf{x}) = \hat{\mu}(\mathbf{x})/Z$ in $\boldsymbol{R}^d$ where $Z$ is the normalization constant. Many sampling algorithms rely on constructing a stochastic process that drives the random particles from an initial distribution $\nu$ that is easy to sample from, to the target distribution $\mu$.

In the variational inference framework, one seeks to construct a parameterized stochastic process to achieve this goal. Denote by $\Omega = C([0,T]; \boldsymbol{R}^d)$ the path space consisting of all possible trajectories and by $\mathcal{P}$ the measure over $\Omega$ induced by a stochastic process with terminal distribution $\mu$ at time $T$. Let $\mathcal{Q}$ be the measure induced by a parameterized stochastic and denote its marginal distribution at $T$ by $\mu^{\mathcal{Q}}$. Then, by the data processing inequality, the Kullback-Leibler divergence (KL) between marginal distributions $\mu^Q$ and $\mu$ can be bounded by

$$D_{\mathrm{KL}}(\mu^{\mathcal{Q}} \| \mu) \leq D_{\mathrm{KL}}(\mathcal{Q} \| \mathcal{P}) := \int_{\Omega} \mathrm{d}\mathcal{Q} \log \frac{\mathrm{d}\mathcal{Q}}{\mathrm{d}\mathcal{P}}. \tag{1}$$

Thus, $D_{\mathrm{KL}}(\mathcal{Q} \| \mathcal{P})$ serves as a performance metric for the sampler, and a small $D_{\mathrm{KL}}(\mathcal{Q} \| \mathcal{P})$ value corresponds to a good sampler.

### 2.2 STOCHASTIC CONTROL

Consider a model characterized by a special stochastic differential equation (SDE) (Särkkä & Solin, 2019)

$$\mathrm{d}\mathbf{x}_t = \mathbf{u}_t \mathrm{d}t + \mathrm{d}\mathbf{w}_t, \ \mathbf{x}_0 \sim \nu, \tag{2}$$

where $\mathbf{x}_t$, $\mathbf{u}_t$ denote state and control input respectively, and $\mathbf{w}_t$ denotes standard Brownian motion. In stochastic control, the goal is to find an feedback control strategy that minimizes a certain given cost function.

The standard stochastic control problem can be associated with any cost and any dynamics. In this work, we only consider cost of the form

$$\mathbb{E} \left[ \int_0^T \frac{1}{2} \|\mathbf{u}_t\|^2 \, \mathrm{d}t + \Psi(\mathbf{x}_T) \mid \mathbf{x}_0 \sim \nu \right], \tag{3}$$

where $\Psi$ represents the terminal cost. The corresponding optimal control problem can be solved via dynamic programming (Bertsekas et al., 2000), which amounts to solving the Hamilton-Jacobi-Bellman (HJB) equation (Evans, 1998)

$$\frac{\partial V_t}{\partial t} - \frac{1}{2} \nabla V_t' \nabla V_t + \frac{1}{2} \Delta V_t = 0, \ \ V_T(\cdot) = \Psi(\cdot). \tag{4}$$

The space-time function $V_t(\mathbf{x})$ is known as *cost-to-go* function or *value function*. The optimal policy can be computed from $V_t(\mathbf{x})$ as (Pavon, 1989)

$$\mathbf{u}_t^*(\mathbf{x}) = -\nabla V_t(\mathbf{x}). \tag{5}$$

## 3 PATH INTEGRAL SAMPLER

It turns out that, with a proper choice of initial distribution $\nu$ and terminal loss function $\Psi$, the stochastic control problem coincides with sampling problem, and the optimal policy drives samples from $\nu$ to $\mu$ perfectly. The process under optimal control can be viewed as the posterior of uncontrolled dynamics conditioned on target distribution as illustrated in Fig 1. Throughout, we denote by $\mathcal{Q}^u$ the path measure associated with control policy $\mathbf{u}$. We also denote by $\mu^0$ the terminal distribution of the uncontrolled process $\mathcal{Q}^0$. For the ease of presentation, we begin with sampling from a normalized density $\mu$, and then generalize the results to unnormalized $\hat{\mu}$ in Section 3.4.

### 3.1 PATH INTEGRAL AND VALUE FUNCTION

Thanks to the special cost structure, the nonlinear HJB eq (4) can be transformed into a linear partial differential equation (PDE)

$$\frac{\partial \phi_t}{\partial t} + \frac{1}{2}\Delta\phi_t = 0, \ \ \phi_T(\cdot) = \exp\{-\Psi(\cdot)\} \tag{6}$$

by logarithmic transformation (Särkkä & Solin, 2019) $V_t(\mathbf{x}) = -\log\phi_t(\mathbf{x})$. By the celebrated Feynman-Kac formula (Øksendal, 2003), the above has solution

$$\phi_t(\mathbf{x}) = \mathbb{E}_{\mathcal{Q}^0}[\exp(-\Psi(\mathbf{x}_T))|\mathbf{x}_t = \mathbf{x}]. \tag{7}$$

We remark that eq (7) implies that the optimal value function can be evaluated without knowing the optimal policy since the above expectation is with respect to the uncontrolled process $\mathcal{Q}^0$. This is exactly the Path Integral control theory (Theodorou et al., 2010; Theodorou & Todorov, 2012; Thijssen & Kappen, 2015). Furthermore, the optimal control at $(t, \mathbf{x})$ is

$$\mathbf{u}_t^*(\mathbf{x}) = \nabla\log\phi_t(\mathbf{x}) = \lim_{s\searrow t}\frac{\mathbb{E}_{\mathcal{Q}^0}\{\exp\{-\Psi(\mathbf{x}_T)\}\int_t^s d\mathbf{w}_t \mid \mathbf{x}_t = \mathbf{x}\}}{(s-t)\mathbb{E}_{\mathcal{Q}^0}\{\exp\{-\Psi(\mathbf{x}_T)\} \mid \mathbf{x}_t = \mathbf{x}\}}, \tag{8}$$

meaning that $\mathbf{u}_t^*(\mathbf{x})$ can also be estimated by uncontrolled trajectories.

### 3.2 SAMPLING AS A STOCHASTIC OPTIMAL CONTROL PROBLEM

There are infinite choices of control strategy $\mathbf{u}$ such that eq (2) has terminal distribution $\mu$. We are interested in the one that minimizes the KL divergence to the prior uncontrolled process. This is exactly the Schrödinger bridge problem (Pavon, 1989; Dai Pra, 1991; Chen et al., 2016; 2021), which has been shown to have a stochastic control formulation with cost being control efforts. In cases where $\nu$ is a Dirac distribution, it is the same as the stochastic control problem in Section 2.2 with a proper terminal cost as characterized in the following result (Tzen & Raginsky, 2019).

**Theorem 1** (Proof in appendix A). *When $\nu$ is a Dirac distribution and terminal loss is chosen as* $\Psi(\mathbf{x}_T) = \log\frac{\mu^0(\mathbf{x}_T)}{\mu(\mathbf{x}_T)}$, *the distribution $\mathcal{Q}^*$ induced by the optimal control policy is*

$$\mathcal{Q}^*(\tau) = \mathcal{Q}^0(\tau|\mathbf{x}_T)\mu(\mathbf{x}_T). \tag{9}$$

*Moreover, $\mathcal{Q}^*(\mathbf{x}_T) = \mu(\mathbf{x}_T)$.*

To gain more insight, consider the KL divergence

$$D_{\mathrm{KL}}(\mathcal{Q}^u(\tau)\|\mathcal{Q}^0(\tau|\mathbf{x}_T)\mu(\mathbf{x}_T)) = D_{\mathrm{KL}}(\mathcal{Q}^u(\tau)\|\mathcal{Q}^0(\tau)\frac{\mu(\mathbf{x}_T)}{\mu^0(\mathbf{x}_T)}) = D_{\mathrm{KL}}(\mathcal{Q}^u\|\mathcal{Q}^0) + \mathbb{E}_{\mathcal{Q}^u}[\log\frac{\mu^0}{\mu}]. \tag{10}$$

Thanks to the Girsanov theorem (Särkkä & Solin, 2019),

$$\frac{d\mathcal{Q}^u}{d\mathcal{Q}^0} = \exp(\int_0^T \frac{1}{2}\|\mathbf{u}_t\|^2 dt + \mathbf{u}_t'd\mathbf{w}_t). \tag{11}$$

It follows that

$$D_{\mathrm{KL}}(\mathcal{Q}^u\|\mathcal{Q}^0) = \mathbb{E}_{\mathcal{Q}^u}[\int_0^T \frac{1}{2}\|\mathbf{u}_t\|^2 dt]. \tag{12}$$

Plugging eq (12) into eq (10) yields

$$D_{\mathrm{KL}}(\mathcal{Q}^u(\tau)\|\mathcal{Q}^0(\tau|\mathbf{x}_T)\mu(\mathbf{x}_T)) = \mathbb{E}_{\mathcal{Q}^u}[\int_0^T \frac{1}{2}\|\mathbf{u}_t\|^2 dt + \log\frac{\mu^0(\mathbf{x}_T)}{\mu(\mathbf{x}_T)}], \tag{13}$$

which is exactly the cost defined in eq (3) with $\Psi = \log\frac{\mu^0}{\mu}$. Theorem 1 implies that once the optimal control policy that minimizes this cost is found, it can also drive particles from $\mathbf{x}_0 \sim \nu$ to $\mathbf{x}_T \sim \mu$.

---

**Algorithm 1** Training

---

    **Input:** Vector: $\mathbf{x}_0 = 0$, Scalar: $y_0 = 0$
    **Output:** $\mathbf{u}_t(\mathbf{x})$ parameterized by $\theta$
    **Define:** SDE drift $\mathbf{f}(t, [\mathbf{x}_t, y_t]) = [\mathbf{u}_{\theta t}(\mathbf{x}_t), \frac{1}{2} \|\mathbf{u}_{\theta t}(\mathbf{x}_t)\|^2]$, diffusion $\mathbf{g}(t, [\mathbf{x}_t, y_t]) = [1, 0]$
    **loop** epochs
        $\mathbf{x}_T, y_T = \text{sdeint}(\mathbf{f}, \mathbf{g}, [\mathbf{x}_0, y_0], [0, T])$     # Integrate SDE from 0 to $T$ with Neural SDE
        Gradient descent step $\nabla_\theta [y_T + \log \frac{\mu^0(\mathbf{x}_T)}{\mu(\mathbf{x}_T)}]$     # Optimize control policy
    **done**

---

### 3.3 OPTIMAL CONTROL POLICY AND SAMPLER

**Optimal Policy Representation:** Consider the sampling strategy from a given target density by simulating SDE in eq (2) under optimal control. Even though the optimal policy is characterized by eq (8), only in rare case (Gaussian target distribution) it has an analytic closed-form.

For more general target distributions, we can instead evaluate the value function eq (7) via empirical samples using Monte Carlo. The approach is essentially importance sampling whose proposal distribution is the uncontrolled dynamics. However, this approach has two drawbacks. First, it is known that the estimation variance can be intolerably high when the proposal distribution is not close enough to the target distribution (MacKay, 2003). Second, even if the variance is acceptable, without a good proposal, the required samples size increases exponentially with dimension, which prevents the algorithm from being used in high or even medium dimension settings (Neal, 2001).

To overcome the above shortcomings, we parameterize the control policy with a neural network $\mathbf{u}_\theta$. We seek a control policy that minimizes the cost

$$\mathbf{u}^* = \arg\min_{\mathbf{u}} \mathbb{E}_{\mathcal{Q}^u} \left[ \int_0^T \frac{1}{2} \|\mathbf{u}_t\|^2 \, \mathrm{d}t + \log \frac{\mu^0(\mathbf{x}_T)}{\mu(\mathbf{x}_T)} \right]. \tag{14}$$

The formula eq (14) also serves as distance metric between $\mathbf{u}_\theta$ and $\mathbf{u}^*$ as in eq (13).

**Gradient-informed Policy Representation:** It is believed that proper prior information can significantly boost the performance of neural network (Goodfellow et al., 2016). The score $\nabla \log \mu(\mathbf{x})$ has been used widely to improve the proposal distribution in MCMC (Li et al., 2020b; Hoffman & Gelman, 2014) and often leads to better results compared with proposals without gradient information. In the same spirit, we incorporate $\nabla \log \mu(\mathbf{x})$ and parameterize the policy as

$$\mathbf{u}_t(\mathbf{x}) = \text{NN}_1(t, \mathbf{x}) + \text{NN}_2(t) \times \nabla \log \mu(\mathbf{x}), \tag{15}$$

where $\text{NN}_1$ and $\text{NN}_2$ are two neural networks. Empirically, we also found that the gradient information leads to faster convergence and smaller discrepancy $D_{\text{KL}}(\mathcal{Q}^u \| \mathcal{Q}^*)$. We remark that PIS with policy eq (15) can be viewed as a modulated Langevin dynamics (MacKay, 2003) that achieves $\mu$ within finite time $T$ instead of infinite time.

**Optimize Policy:** Optimizing $\mathbf{u}_\theta$ requires the gradient of loss in eq (14), which involves $\mathbf{u}_t$ and the terminal state $\mathbf{x}_T$. To calculate gradients, we rely on backpropagation through trajectories. We train the control policy with recent techniques of Neural SDEs (Li et al., 2020a; Kidger et al., 2021), which greatly reduce memory consumption during training. The gradient computation for Neural SDE is based on stochastic adjoint sensitivity, which generalizes the adjoint sensitivity method for Neural ODE (Chen et al., 2018). Therefore, the backpropagation in Neural SDE is another SDE associated with adjoint states. Unlike the training of traditional deep MLPs which often runs into gradient vanishing/exploding issues, the training of Neural SDE/ODE is more stable and not sensitive the number of discretization steps (Chen et al., 2018; Kidger et al., 2021). We augment the origin SDE with state $\int_0^t \frac{1}{2} \|\mathbf{u}_s\|^2 \, \mathrm{d}s$ such that the whole training can be conducted end to end. The full training procedure in provided in Algorithm 1.

**Wasserstein distance bound:** The PIS trained by Algorithm 1 can not generate unbiased samples from the target distribution $\mu$ for two reasons. First, due to the non-convexity of networks and randomness of stochastic gradient descent, there is no guarantee that the learned policy is optimal. Second, even if the learned policy is optimal, the time-discretization error in simulating SDEs is in-

evitable. Fortunately, the following theorem quantifies the Wasserstein distance between the sampler and the target density. (More details and a formal statement can be found in appendix C)

**Theorem 2** (Informal). *Under mild condition, with sampling step size $\Delta t$, if $\|\mathbf{u}_t^* - \mathbf{u}_t\|^2 \leq d\epsilon$ for any $t$, then*

$$W_2(\mathcal{Q}^u(\mathbf{x}_T), \mu(\mathbf{x}_T)) = \mathcal{O}(\sqrt{Td(\Delta t + \epsilon)}). \tag{16}$$

### 3.4 IMPORTANCE SAMPLING

The training procedure for PIS does not guarantee its optimality. To compensate for the mismatch between the trained policy and the optimal policy, we introduce importance weight to calibrate generated samples. The importance weight can be calculated by (more details in appendix B)

$$w^u(\tau) = \frac{\mathrm{d}\mathcal{Q}^*(\tau)}{\mathrm{d}\mathcal{Q}^u(\tau)} = \exp(\int_0^T -\frac{1}{2}\|\mathbf{u}_t\|^2 \,\mathrm{d}t - \mathbf{u}_t'\mathrm{d}\mathbf{w}_t - \Psi(\mathbf{x}_T)). \tag{17}$$

We note eq (17) resembles training objective eq (14). Indeed, eq (14) is the average of logarithm of eq (17). If the trained policy is optimal, that is, $\mathcal{Q}^u = \mathcal{Q}^*$, all the particles share the same weight. We summarize the sampling algorithm in Algorithm 2.

**Effective Sample Size:** The Effective Sample Size (ESS), $\mathrm{ESS}^u = \frac{1}{\mathbb{E}_{\mathcal{Q}^u}[(w^u)^2]}$, is a popular metric to measure the variance of importance weights. ESS is often accompanied by resampling trick (Tokdar & Kass, 2010) to mitigate deterioration of sample quality. ESS

---

**Algorithm 2** Sampling

**Input:** Vector: $\mathbf{x}_0 = 0$, Scalar: $y_0 = 0$
**Output:** Samples with weights
**for** $i \leftarrow 1$ to $N$ **do**
$\quad \Delta t = t_i - t_{i-1}, \Delta\mathbf{w} \sim \mathcal{N}(0, \Delta t\mathcal{I}),$
$\quad \mathbf{x}_i = \mathbf{x}_{i-1} + \mathbf{u}\Delta t + \Delta\mathbf{w}$
$\quad y_i = y_{i-1} + \mathbf{u}'\Delta\mathbf{w} + \frac{1}{2}\|\mathbf{u}\|^2 \Delta t$
**end for**
**Outputs:**
$\quad \mathbf{x}_N, \exp(-y_N - \log\frac{\mu^0(\mathbf{x}_N)}{\mu(\mathbf{x}_N)})$

---

is also regarded as a metric for quantifying goodness of sampler based on importance sampling. Low ESS means that estimation or downstream tasks based on such sampling methods may suffer from a high variance. ESS of most importance samplers is decreasing along the time. Thanks to the adaptive control policy in PIS, we can quantify the ESS of PIS based on the optimality of learned policy. For the sake of completeness, the proof in provided in appendix D.

**Theorem 3** (Corollary 7 (Thijssen & Kappen, 2015)). *If $\max_{t,\mathbf{x}}\|\mathbf{u}_t(\mathbf{x}) - \mathbf{u}_t^*(\mathbf{x})\|^2 \leq \frac{\epsilon}{T}$, then*

$$\frac{1}{\mathbb{E}_{\mathcal{Q}^u}[(w^u)^2]} \geq 1 - \epsilon.$$

**Estimation of normalization constants:** In most sampling problems we only have access to the target density up to a normalization constant, denoted by $\hat{\mu} = Z\mu$. PIS can still generate samples following the same protocol with new terminal cost $\hat{\Psi} = \log\frac{\mu^0}{\hat{\mu}} = \Psi - \log Z$. The additional constant $-\log Z$ is independent of $\mathbf{x}_T$ and thus does not affect the optimal policy and the optimization of $\mathbf{u}_\theta$. As a byproduct, we can estimate the normalization constants (more details in appendix E).

**Theorem 4.** *For any given policy $\mathbf{u}$, the logarithm of normalization constant is bounded below by*

$$\mathbb{E}_{\tau\sim\mathcal{Q}^u}[-\hat{S}^u(\tau)] \leq \log Z, \tag{18}$$

*where $\hat{S}^u(\tau) = \int_0^T \frac{1}{2}\|\mathbf{u}_t(\mathbf{x}_t)\|^2 \,\mathrm{d}t + \mathbf{u}_t'(\mathbf{x}_t)\mathrm{d}\mathbf{w}_t + \hat{\Psi}(\mathbf{x}_T)$. The equality holds only when $\mathbf{u} = \mathbf{u}^*$. Moreover, for any sub-optimal policy, an unbiased estimation of $Z$ using importance sampling is*

$$Z = \mathbb{E}_{\tau\sim\mathcal{Q}^u}[\exp(-\hat{S}^u(\tau))]. \tag{19}$$

## 4 EXPERIMENTS

In this section we present empirical evaluations of PIS and the comparisons to several baselines. We also provide details of practical implementations. Inspired by Arbel et al. (2021), we conduct experiments for tasks of Bayesian inference and normalization constant estimation.

We consider three types of relevant methods. The first category is gradient-guided MCMC methods without the annealing trick. It includes the Hamiltonian Monte Carlo (HMC) (MacKay, 2003) and

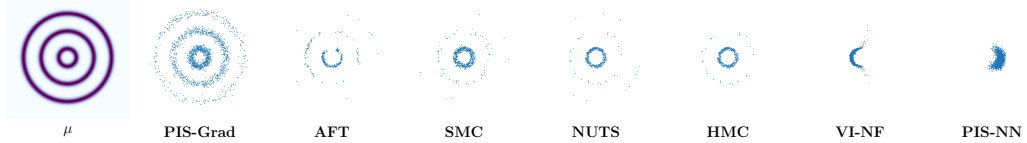

| $\mu$ | PIS-Grad | AFT | SMC | NUTS | HMC | VI-NF | PIS-NN |

Figure 2: Sampling performance on rings-shape density function with 100 steps. The gradient information can help PIS-Grad and MCMC algorithm improve sampling performance.

No-U-Turn Sampler (NUTS) (Hoffman & Gelman, 2014). The second is Sequential Monte Carlo with annealing trick (SMC), which is regarded as state-of-the-art sampling algorithm (Del Moral et al., 2006) in terms of sampling quality. We choose a standard instance of SMC samplers and the recently proposed Annealed Flow Transport Monte Carlo (AFT) (Arbel et al., 2021). Both use a default 10 temperature levels with a linear annealing scheme. We note that there are optimized SMC variants that achieve better performance (Del Moral et al., 2006; Chopin & Papaspiliopoulos, 2020; Zhou et al., 2016). Since the introduction of advanced tricks, we exclude the comparison with those variants for fair comparison purpose. We note PIS can also be augmented with annealing trick, possible improvement for PIS can be explored in the future. Last, the variational normalizing flow (VI-NF) (Rezende & Mohamed, 2015) is also included for comparison. We note that another popular line of sampling algorithms use Stein-Variational Gradient Descent (SVGD) or other particle-based variational inference approaches (Liu & Wang, 2016; Liu et al., 2019). We include the comparison and more discussions on SGVD in appendix F.3 due to its significant difference. In our experiments, the number of steps $N$ of MCMC algorithms and the number of SDE time-discretization steps for PIS work as a proxy for benchmarking computation times.

We also investigate the effects of two different network architectures for Path Integral Sampler. The first one is a time-conditioned neural network without any prior information, which we denote as *PIS-NN*, while the second one incorporates the gradient information of the given energy function as in eq (15), denoted as *PIS-Grad*. When we have an analytical form for the ground truth optimal policy, the policy is denoted as *PIS-GT*. The subscript *RW* is to distinguish PIS with path integral importance weights eq (17) that use eq (19) to estimate normalization constants from the ones without importance weights that use the bound in eq (18) to estimate $Z$. For approaches without the annealing trick, we take default $N = 100$ unless otherwise stated. With annealing, $N$ steps are the default for each temperature level, thus AFT and SMC rougly use 10 times more steps compared with HMC and PIS. We include more details about hyperparameters, training time, sampling efficiency, and more experiments with large $N$ in appendices F and G.

## 4.1 PIS-GRAD VS PIS-NN: IMPORTANCE OF GRADIENT GUIDANCE

We observed that the advantage of PIS-Grad over PIS-NN is clearer when the target density has multiple modes as in the toy example shown in Fig 2. The objective $D_{\mathrm{KL}}(\mathcal{Q}\|\mathcal{Q}^*)$ is known to have *zero forcing*. In particular, when the modes of the density are well separated and $\mathcal{Q}$ is not expressive enough, minimizing $D_{\mathrm{KL}}(\mathcal{Q}\|\mathcal{Q}^*)$ can drive $\mathcal{Q}(\tau)$ to zero on some area, even if $\mathcal{Q}^*(\tau) > 0$ (Fox & Roberts, 2012). PIS-NN and VI-NF generate very similar samples that almost cover half the inner ring. The training objective function of VI-NF can also be viewed as minimizing KL divergence between two trajectory distributions (Wu et al., 2020). The added noise during the process can encourage exploration but it is unlikely such noise only can overcome the local minima. On the other hand, the gradient information can help cover more modes and provide exploring directions.

## 4.2 BENCHMARKING DATASETS

**Mode-separated mixture of Gaussian:** We consider the mixture of Gaussian in 2-dimension. We notice that when the Gaussian modes are not far away from each other, all methods work well. However, when we reduce the variances of the Gaussian distributions and separate the modes of Gaussian, the advantage of PIS becomes clear even in this low dimension task. We generate 2000 samples from each method and plot their kernel density estimate (KDE) in Fig 4. PIS generates samples that are visually indistinguishable from the target density.

**Funnel distribution:** We consider the popular testing distribution in MCMC literature (Hoffman & Gelman, 2014; Hoffman et al., 2019), the 10-dimensional Funnel distribution charaterized by

$$x_0 \sim \mathcal{N}(0, 9), \quad x_{1:9}|x_0 \sim \mathcal{N}(0, \exp(x_0)\mathbf{I}).$$

This distribution can be pictured as a funnel - with $x_0$ wide at the mouth of funnel, getting smaller as the funnel narrows.

| | MG ($d = 2$) | | | Funnel ($d = 10$) | | | LGCP ($d = 1600$) | | |
|---|---|---|---|---|---|---|---|---|---|
| | B | S | $A$ | B | S | $A$ | B | S | $A$ |
| $\text{PIS}_{RW}$-GT | -0.012 | 0.013 | 0.018 | - | - | - | - | - | - |
| PIS-NN | -1.691 | 0.370 | 1.731 | -0.098 | **5e-3** | 0.098 | -92.4 | 6.4 | 92.62 |
| PIS-Grad | -0.440 | 0.024 | 0.441 | -0.103 | 9e-3 | 0.104 | -13.2 | 3.21 | 13.58 |
| $\text{PIS}_{RW}$-NN | -1.192 | 0.482 | 1.285 | -0.018 | 7e-3 | 0.02 | -60.8 | 4.81 | 60.99 |
| $\text{PIS}_{RW}$-Grad | **-0.021** | **0.030** | **0.037** | **-0.008** | 9e-3 | **0.012** | **-1.94** | **0.91** | **2.14** |
| AFT | -0.509 | 0.24 | 0.562 | -0.208 | 0.193 | 0.284 | -3.08 | 1.59 | 3.46 |
| SMC | -0.362 | 0.293 | 0.466 | -0.216 | 0.157 | 0.267 | -435 | 14.7 | 436 |
| NUTS | -1.871 | 0.527 | 1.943 | -0.835 | 0.257 | 0.874 | -1.3e3 | 8.01 | 1.3e3 |
| HMC | -1.876 | 0.527 | 1.948 | -0.835 | 0.257 | 0.874 | -1.3e3 | 8.01 | 1.3e3 |
| VI-NF | -1.632 | 0.965 | 1.896 | -0.236 | 0.0591 | 0.243 | -77.9 | 5.6 | 78.2 |

Table 1: Benchmarking on mode separated mixture of Gaussian (MG), Funnel distribution and Log Gaussian Cox Process (LGCP) for estimation log normalization constants. *B* and *S* stand for estimation bias and standard deviation among 100 runs and $A^2 = B^2 + S^2$.

**Log Gaussian Cox Process:** We further investigate the normalization constant estimation problem for the challenging log Gaussian Cox process (LGCP), which is designed for modeling the positions of Finland pine saplings. In LGCP (Salvatier et al., 2016), an underlying field $\lambda$ of positive real values is modeled using an exponentially-transformed Gaussian process. Then $\lambda$ is used to parameterize Poisson points process to model locations of pine saplings. The posterior density is

$$\lambda(\mathbf{x}) \sim \exp\left(-\frac{(\mathbf{x} - \mu)^T K^{-1}(\mathbf{x} - \mu)}{2}\right) \prod_{i \in d} \exp(x_i y_i - \alpha \exp x_i), \tag{20}$$

where $d$ denotes the size of discretized grid and $y_i$ denotes observation information. The modeling parameters, including normal distribution and $\alpha$, follow Arbel et al. (2021) (See appendix F).

Tab 1 clearly shows the advantages of PIS for the above three datasets, and supports the claim that importance weight helps improve the estimation of log normalization constants, based on the comparison between $\text{PIS}_{RW}$ and PIS. We also found that PIS-Grad trained with gradient information outperforms PIS-NN. The difference is more obvious in datasets that have well-separated modes, such as MG and LGCP, and less obvious on unimodal distributions like Funnel.

In all cases, $\text{PIS}_{RW}$-Grad is better than AFT and SMC. Interestingly, even *without* annealing and gradient information of target density, $\text{PIS}_{RW}$-NN can outperform SMC with annealing trick and HMC kernel for the Funnel distribution.

### 4.3 ADVANTAGE OF THE SPECIALIZED SAMPLING ALGORITHM

From the perspective of particles dynamics, most existing MCMC algorithms are invariant to the target distribution. Therefore, particles are driven by gradient and random noise in a way that is independent of the given target distribution. In contrast, PIS learns different strategies to combine gradient information and noise for different target densities. The specialized sampling algorithm can generate samples more efficiently and shows better performance empirically in our experiments. The advantage can be showed in various datasets, from unimodal distributions like the Funnel distribution to multimodal distributions. The benefits and efficiency of PIS are more obvious in high dimensional settings as we have shown.

### 4.4 ALANINE DIPEPTIDE

Building on the success achieved by flow models in the generation of asymptotically unbiased samples from physics models (LeCun, 1998), we investigate the applications in the sampling of molec-

| KL* | $\mu$ | $\phi$ | $\eta_1$ | $\psi$ | $\eta_2$ | $\eta_3$ |
|---|---|---|---|---|---|---|
| VI-NF | 175.6 $\pm$4.5 | 24.2$\pm$ 4.1 | 3.1 $\pm$ 0.05 | 14.6$\pm$ 6.4 | 7e-2$\pm$5e-3 | **8.5e-2$\pm$3.5e-3** |
| SMC | 183.3 $\pm$2.3 | 18.3$\pm$ 2.1 | 0.32 $\pm$ 0.08 | 9.6 $\pm$ 1.2 | 0.12$\pm$0.05 | 0.15 $\pm$ 9e-3 |
| SNF | 181.8 $\pm$0.75 | 6.3$\pm$ 0.71 | **0.17$\pm$0.05** | 1.58 $\pm$ 0.36 | 0.11$\pm$ 0.03 | 8.8e-2 $\pm$8e-3 |
| PIS-NN | **171.3 $\pm$0.61** | **5.2$\pm$ 0.35** | 0.32$\pm$0.03 | **1.03 $\pm$ 0.23** | **5e-2$\pm$5e-3** | 8.7e-2$\pm$3e-3 |

Table 2: KL-divergences comparison among variational approaches of generated density with target density in overall atom states distribution and five multimodal torsion angles. We emphasis KL* denote the KL divergence between unnormalized distribution due to lack of ground truth normalization constants. Mean and standard deviation are conducted with five different random seeds.

ular structure from a simulation of Alanine dipeptide as introduced in Wu et al. (2020). The target density of molecule is $\hat{\mu} = \exp(-E(\mathbf{x}_{[0:65]}) - \frac{1}{2} \left\| \mathbf{x}_{[66:131]} \right\|^2)$.

We compare PIS with popular variational approaches used in generating samples from the above model. More specifically, we consider VI-NF, and Stochastic Normalizing Flow (SNF) (Wu et al., 2020). SNF is very close to AFT (Arbel et al., 2021). Both of them couple deterministic normalizing flow layers and MCMC blocks except SNF uses an amortized structure. We include more details of MCMC kernel and modification in appendix F. We show a generated molecular in Fig 3 and quantitative comparison in terms of KL divergence in Tab 2, including overall atom states distribution and five multimodal torsion angles (backbone angles $\phi, \psi$ and methyl rotation angles $\eta_1, \eta_2, \eta_3$). We remark that unweighted samples are used to approximate the density of torsion angles and all approaches do not use gradient information. Clearly, PIS gives lower divergence.

Figure 3: Sampled Alanine dipeptide molecules

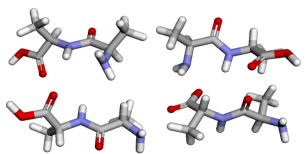

### 4.5 Sampling in Variational Autoencoder latent space

In this experiment we investigate sampling in the latent space of a trained Variational Autoencoder (VAE). VAE aims to minimize $D_{\mathrm{KL}}(q(\mathbf{x})q_\phi(\mathbf{z}|\mathbf{x})\|p(\mathbf{z})p_\theta(\mathbf{x}|\mathbf{z}))$, where $q_\phi(\mathbf{z}|\mathbf{x})$ represents encoder and $p_\theta$ for a decoder with latent variable $\mathbf{z}$ and data $\mathbf{x}$. We investigate the posterior distribution

$$\mathbf{z} \sim p(\mathbf{z})p_\theta(\mathbf{x}|\mathbf{z}). \tag{21}$$

The normalization constant of such target unnormalized density function $p(\mathbf{z})p_\theta(\mathbf{x}|\mathbf{z})$ is exactly the likelihood of data points $p_\theta(\mathbf{x})$, which serves as an evaluation metric for the trained VAE.

We investigate a vanilla VAE model trained with plateau loss on the binary MNIST (LeCun, 1998) dataset. For each distribution, we regard the average estimation from 10 long-run SMC with 1000 temperature levels as the ground truth normalization constant. We choose 100 images randomly and run the various approaches on estimating normalization of those posterior distributions in eq (21) and report the average performance in Tab 3. PIS has a lower bias and variance.

Table 3: Estimation of $\log p_\theta(x)$ of a trained VAE.

| | B | S | $\sqrt{\mathrm{B}^2 + \mathrm{S}^2}$ |
|---|---|---|---|
| VI-NF | -2.3 | 0.76 | 2.42 |
| AFT | -1.7 | 0.95 | 1.96 |
| SMC | -10.6 | 2.01 | 10.79 |
| $\mathrm{PIS}_{RW}$-NN | -1.9 | 0.81 | 2.06 |
| $\mathrm{PIS}_{RW}$-Grad | **-0.87** | **0.31** | **0.92** |

## 5 Conclusion

**Contributions.** In this work, we proposed a new sampling algorithm, Path Integral Sampler, based on the connections between sampling and stochastic control. The control can drive particles from a simple initial distribution to a target density perfectly when the policy is optimal for an optimal control problem whose terminal cost depends on the target distribution. Furthermore, we provide a calibration based on importance weights, ensuring sampling quality even with sub-optimal policies.

**Limitations.** Compared with most popular non-learnable MCMC algorithms, PIS requires training neural networks for the given distributions, which adds additional computational overhead, though this can be mitigated with amortization. Besides, the sampling quality of PIS in finite steps depends on the optimality of trained network. Improper choices of hyperparameters may lead to numerical issues and failure modes as discussed in appendix G.1.

## 6 Reproducibility Statement

The detailed discussions on assumptions and proofs of theorems presented in the main paper are included in appendices A and C to E. The training settings and implementation tips of the algorithms are included in appendices F and G. An implementation based on PyTorch (Paszke et al., 2019) of PIS can be found in `https://github.com/qsh-zh/pis`.

### Acknowledgments

The authors would like to thank the anonymous reviewers for useful comments. This work is partially supported by NSF ECCS-1942523 and NSF CCF-2008513.

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

# A    PROOF OF THEOREM 1

Before proving our main theorem, we introduce the following important lemma.

**Lemma 4.1** (Dai Pra (1991); Pavon (1989)). *The transition density associated with optimal control policy $\mathbf{u}^*$ for eq (2) and eq (3) is*

$$Q_{s,t}^*(\mathbf{x}, \mathbf{y}) = Q_{s,t}^0(\mathbf{x}, \mathbf{y}) \frac{\phi_t(\mathbf{y})}{\phi_s(\mathbf{x})}, \tag{22}$$

*where $Q_{s,t}^u(\mathbf{x}, \mathbf{y})$ denotes the transition probability from state $\mathbf{x}$ at time $s$ to state $\mathbf{y}$ at time $t$.*

**Proof of Theorem 1:** We denote the initial Dirac distribution by $\nu = \delta_{\bar{\mathbf{x}}_0}$. Combining eq (7) and $V_t(\mathbf{x}) = -\log \phi_t(\mathbf{x})$ we obtain

$$V_0(\bar{\mathbf{x}}_0) = -\log E_{Q^0}[\exp(-\Psi(\mathbf{x}_T))] = -\log(\int \frac{\mu}{\mu^0} d\mu^0) = 0.$$

Therefore, we can evaluate the KL divergence between $Q^*$ and $Q^0(\tau|\mathbf{x}_T)\mu(\mathbf{x}_T)$ as

$$D_{\mathrm{KL}}(Q^*(\tau) \| Q^0(\tau|\mathbf{x}_T)\mu(\mathbf{x}_T)) = \mathbb{E}_{\tau \sim Q^*}\left[\int_0^T \frac{1}{2}\|\mathbf{u}_t^*\|^2 \, dt + \Psi(\mathbf{x}_T)\right] = V_0(\bar{\mathbf{x}}_0) = 0.$$

The first equality is based on eq (13). Next, we show that $\mathbf{x}_T^* \sim \mu$. The above equations imply $Q_{0,T}^*(\bar{\mathbf{x}}_0, \mathbf{y})d\mathbf{y} = \exp(-\Psi(\mathbf{y}))\mu^0(d\mathbf{y})$. It follows that

$$\mathbb{P}[\mathbf{x}_T^* \in A] = \int_A Q_{0,T}^*(\bar{\mathbf{x}}_0, \mathbf{y})d\mathbf{y} = \int_A \exp(-\Psi(\mathbf{y}))\mu^0(d\mathbf{y}) = \mu(A).$$

# B    PROOF OF IMPORTANCE WEIGHTS

By definition

$$w^u(\tau) = \frac{dQ^*(\tau)}{dQ^u(\tau)} = \frac{dQ^*(\tau)}{dQ^0(\tau)}\frac{dQ^0(\tau)}{dQ^u(\tau)}. \tag{23}$$

Plugging eq (11) and eq (22) into the above we obtain

$$w^u(\tau) = \exp\left(\int_0^T -\frac{1}{2}\|\mathbf{u}_t\|^2 \, dt - \mathbf{u}_t' d\mathbf{w}_t - \log \frac{\mu^0(\mathbf{x}_T)}{\mu(\mathbf{x}_T)}\right) = \exp\left(\int_0^T -\frac{1}{2}\|\mathbf{u}_t\|^2 \, dt - \mathbf{u}_t' d\mathbf{w}_t - \Psi(\mathbf{x}_T)\right).$$

# C    PROOF OF THEOREM 2

## C.1    LIPCHITZ CONDITON AND PRELIMINARY LEMMA

To ease the burden of notations, we assume $\mathbf{x}_0 \sim \delta_0$. Our conclusion and proof can be generalized to other Dirac distributions easily. We start by assuming some conditions on the Lipschitz of optimal policy $\mathbf{u}^*$. It promises the existence of a unique strong solution with $\mathbf{x}_T \sim \mu$. The conditions and properties are studied in Schrödinger-Föllmer process, which dates back to the Schrödinger problem. For proof of existence of a unique strong solution and detailed discussion, we refer the reader to Dai Pra (1991); Léonard (2014); Eldan et al. (2020); Huang et al. (2021).

**Condition 1.**

$$\|\mathbf{u}_t^*(\mathbf{x})\|_2^2 \leq C_0(1 + \|\mathbf{x}\|^2) \tag{24}$$

*and*

$$\left\|\mathbf{u}_{t_1}^*(\mathbf{x}_1) - \mathbf{u}_{t_2}^*(\mathbf{x}_2)\right\| \leq C_1(\|\mathbf{x}_1 - \mathbf{x}_2\| + |t_1 - t_2|^{\frac{1}{2}}). \tag{25}$$

Below is the formal statement of Theorem 2.

**Theorem 5.** *Under Condition 1, with sampling step size $\Delta t$, if $\|\mathbf{u}_t^* - \mathbf{u}_t\|^2 \leq d\epsilon$ for any $t$, then*

$$W_2(Q^u(\mathbf{x}_T), \mu(\mathbf{x}_T)) = \mathcal{O}(\sqrt{Td(\Delta t + \epsilon)}). \tag{26}$$

We introduce the following lemma before stating the proof for Theorem 2.

**Lemma 5.1.** *(Huang et al., 2021, Lemma A.2.) Assume Condition 1 holds, then the following inequality holds for $\mathbf{x}_t$ generated from $\mathbf{u}^*$,*

$$\mathbb{E}[\|\mathbf{x}_{t_2} - \mathbf{x}_{t_1}\|^2] \leq 4C_0 \exp(2C_0)(C_0 + d)(t_2 - t_1)^2 + 2C_0(t_2 - t_1)^2 + 2d|t_2 - t_1|, \ t_1, t_2 \in [0, T]. \tag{27}$$

### C.2  PROOF OF THEOREM 2

We denote by $\mathbf{x}^*_{(0:T)}$ the trajectory controlled by the optimal policy $\mathbf{u}^*$, and by $\{\mathbf{x}_{t_0}, \mathbf{x}_{t_1}, \cdots, \mathbf{x}_{t_N}\}$ the discrete time process with sub-optimal policy $\mathbf{u}$ over discrete time $\{t_k\}$ such that $t_0 = 0, t_N = T$ and $t_k - t_{k-1} = \Delta t$. The process $\{\mathbf{x}_{t_k}\}$ can be extended to continuous time setting as

$$\mathbf{x}_{t_k} = \mathbf{x}_{t_{k-1}} + \int_{t_{k-1}}^{t_k} \mathbf{u}_{t_{k-1}}(\mathbf{x}_{t_{k-1}}) ds + d\mathbf{w}_s.$$

The key of our proof is the bound of $\left\|\mathbf{x}_{t_k} - \mathbf{x}^*_{t_k}\right\|^2$ as

$$\left\|\mathbf{x}_{t_k} - \mathbf{x}^*_{t_k}\right\|^2 = \left\|\mathbf{x}_{t_{k-1}} + \int_{t_{k-1}}^{t_k} \mathbf{u}_{t_{k-1}}(\mathbf{x}_{t_{k-1}}) ds + d\mathbf{w}_s - [\mathbf{x}^*_{t_{k-1}} + \int_{t_{k-1}}^{t_k} \mathbf{u}^*_s(\mathbf{x}^*_s) ds + d\mathbf{w}_s]\right\|^2$$

$$\leq \left\|\mathbf{x}_{t_{k-1}} - \mathbf{x}^*_{t_{k-1}}\right\|^2 + \left\|\int_{t_{k-1}}^{t_k} [\mathbf{u}^*_s(\mathbf{x}^*_s) - \mathbf{u}_{t_{k-1}}(\mathbf{x}_{t_{k-1}})] ds\right\|^2$$

$$+ 2\left\|\mathbf{x}_{t_{k-1}} - \mathbf{x}^*_{t_{k-1}}\right\| \left\|\int_{t_{k-1}}^{t_k} [\mathbf{u}^*_s(\mathbf{x}^*_s) - \mathbf{u}_{t_{k-1}}(\mathbf{x}_{t_{k-1}})] ds\right\|$$

$$\leq \left\|\mathbf{x}_{t_{k-1}} - \mathbf{x}^*_{t_{k-1}}\right\|^2 + (\int_{t_{k-1}}^{t_k} \left\|\mathbf{u}^*_s(\mathbf{x}^*_s) - \mathbf{u}_{t_{k-1}}(\mathbf{x}_{t_{k-1}})\right\| ds)^2$$

$$+ 2\left\|\mathbf{x}_{t_{k-1}} - \mathbf{x}^*_{t_{k-1}}\right\| (\int_{t_{k-1}}^{t_k} \left\|\mathbf{u}^*_s(\mathbf{x}^*_s) - \mathbf{u}_{t_{k-1}}(\mathbf{x}_{t_{k-1}})\right\| ds)$$

$$\leq (1 + \alpha)\left\|\mathbf{x}_{t_{k-1}} - \mathbf{x}^*_{t_{k-1}}\right\|^2 + (1 + \frac{1}{\alpha})(\int_{t_{k-1}}^{t_k} \left\|\mathbf{u}^*_s(\mathbf{x}^*_s) - \mathbf{u}_{t_{k-1}}(\mathbf{x}_{t_{k-1}})\right\| ds)^2$$

$$\leq (1 + \alpha)\left\|\mathbf{x}_{t_{k-1}} - \mathbf{x}^*_{t_{k-1}}\right\|^2$$

$$+ (1 + \frac{1}{\alpha})(t_k - t_{k-1})(\int_{t_{k-1}}^{t_k} \left\|\mathbf{u}^*_s(\mathbf{x}^*_s) - \mathbf{u}_{t_{k-1}}(\mathbf{x}_{t_{k-1}})\right\|^2 ds), \tag{28}$$

where the first and second inequalities are based on the triangle inequality, the third inequality is based $2ab \leq \alpha a^2 + \frac{1}{\alpha}b^2$ for any $\alpha > 0$, and the forth inequality is based on the Cauchy-Schwarz inequality.

In the following we bound the second term in eq (28) as

$$\left\|\mathbf{u}^*_s(\mathbf{x}^*_s) - \mathbf{u}_{t_{k-1}}(\mathbf{x}_{t_{k-1}})\right\|^2$$

$$= \left\|\mathbf{u}^*_s(\mathbf{x}^*_s) - \mathbf{u}^*_{t_{k-1}}(\mathbf{x}_{t_{k-1}}) + \mathbf{u}^*_{t_{k-1}}(\mathbf{x}_{t_{k-1}}) - \mathbf{u}_{t_{k-1}}(\mathbf{x}_{t_{k-1}})\right\|^2$$

$$\leq (1 + \beta)\left\|\mathbf{u}^*_s(\mathbf{x}^*_s) - \mathbf{u}^*_{t_{k-1}}(\mathbf{x}_{t_{k-1}})\right\|^2 + (1 + \frac{1}{\beta})\left\|\mathbf{u}^*_{t_{k-1}}(\mathbf{x}_{t_{k-1}}) - \mathbf{u}_{t_{k-1}}(\mathbf{x}_{t_{k-1}})\right\|^2$$

$$\leq 2C_1^2(1 + \beta)[\left\|\mathbf{x}^*_s - \mathbf{x}_{t_{k-1}}\right\|^2 + |s - t_{k-1}|] + (1 + \frac{1}{\beta})\left\|\mathbf{u}^*_{t_{k-1}}(\mathbf{x}_{t_{k-1}}) - \mathbf{u}_{t_{k-1}}(\mathbf{x}_{t_{k-1}})\right\|^2,$$

where the first inequality uses $2ab \leq \beta a^2 + \frac{1}{\beta}b^2$ for an arbitrary $\beta > 0$ and the second one is based on eq (25). It follows that

$$\int_{t_{k-1}}^{t_k} \left\| \mathbf{u}_s^*(\mathbf{x}_s^*) - \mathbf{u}_{t_{k-1}}(\mathbf{x}_{t_{k-1}}) \right\|^2 \mathrm{d}s$$

$$\leq \int_{t_{k-1}}^{t_k} 2C_1^2(1+\beta) \left\| \mathbf{x}_s^* - \mathbf{x}_{t_{k-1}} \right\|^2 \mathrm{d}s + \int_{t_{k-1}}^{t_k} 2C_1^2(1+\beta)(s - t_{k-1})\mathrm{d}s$$

$$+ \int_{t_{k-1}}^{t_k} (1 + \frac{1}{\beta}) \left\| \mathbf{u}_{t_{k-1}}^*(\mathbf{x}_{t_{k-1}}) - \mathbf{u}_{t_{k-1}}(\mathbf{x}_{t_{k-1}}) \right\|^2 \mathrm{d}s.$$

Thus, for stepsize $\Delta t = t_k - t_{k-1}$, we establish

$$\int_{t_{k-1}}^{t_k} \left\| \mathbf{u}_s^*(\mathbf{x}_s^*) - \mathbf{u}_{t_{k-1}}(\mathbf{x}_{t_{k-1}}) \right\|^2 \mathrm{d}s$$

$$\leq \int_{t_{k-1}}^{t_k} 2C_1^2(1+\beta) \left\| \mathbf{x}_s^* - \mathbf{x}_{t_{k-1}} \right\|^2 \mathrm{d}s$$

$$+ C_1^2(1+\beta)\Delta t^2 + (1 + \frac{1}{\beta}) \left\| \mathbf{u}_{t_{k-1}}^*(\mathbf{x}_{t_{k-1}}) - \mathbf{u}_{t_{k-1}}(\mathbf{x}_{t_{k-1}}) \right\|^2 \Delta t. \tag{29}$$

Next we bound $\left\| \mathbf{x}_s^* - \mathbf{x}_{t_{k-1}} \right\|^2$ in eq (29) as

$$\left\| \mathbf{x}_s^* - \mathbf{x}_{t_{k-1}} \right\|^2 \leq (1+\eta) \left\| \mathbf{x}_s^* - \mathbf{x}_{t_{k-1}}^* \right\|^2 + (1 + \frac{1}{\eta}) \left\| \mathbf{x}_{t_{k-1}}^* - \mathbf{x}_{t_{k-1}} \right\|^2, \tag{30}$$

where the inequality is based on $(a+b)^2 \leq (1+\eta)a^2 + (1+\frac{1}{\eta})b^2$ for an arbitrary $\eta > 0$.

Plugging eq (30) into eq (29) yields

$$\int_{t_{k-1}}^{t_k} \left\| \mathbf{u}_s^*(\mathbf{x}_s^*) - \mathbf{u}_{t_{k-1}}(\mathbf{x}_{t_{k-1}}) \right\|^2 \mathrm{d}s$$

$$\leq (1 + \frac{1}{\beta}) \left\| \mathbf{u}_{t_{k-1}}^*(\mathbf{x}_{t_{k-1}}) - \mathbf{u}_{t_{k-1}}(\mathbf{x}_{t_{k-1}}) \right\|^2 \Delta t + C_1^2(1+\beta)\Delta t^2$$

$$+ 2C_1^2(1+\beta)(1+\frac{1}{\eta}) \left\| \mathbf{x}_{t_{k-1}}^* - \mathbf{x}_{t_{k-1}} \right\|^2 \Delta t + 2C_1^2(1+\beta)(1+\eta) \int_{t_{k-1}}^{t_k} \left\| \mathbf{x}_s^* - \mathbf{x}_{t_{k-1}}^* \right\|^2 \mathrm{d}s. \tag{31}$$

Plugging eq (29) and (31) into eq (28) yields

$$LHS \leq [1 + \alpha + 2C_1^2(1+\frac{1}{\alpha})(1+\beta)(1+\frac{1}{\eta})\Delta t^2] \left\| \mathbf{x}_{t_{k-1}}^* - \mathbf{x}_{t_{k-1}} \right\|^2$$

$$+ 2C_1^2(1+\frac{1}{\alpha})(1+\beta)(1+\eta)\Delta t \int_{t_{k-1}}^{t_k} \left\| \mathbf{x}_s^* - \mathbf{x}_{t_{k-1}}^* \right\|^2 \mathrm{d}s$$

$$+ (1+\frac{1}{\alpha})(1+\frac{1}{\beta}) \left\| \mathbf{u}_{t_{k-1}}^*(\mathbf{x}_{t_{k-1}}^*) - \mathbf{u}_{t_{k-1}}(\mathbf{x}_{t_{k-1}}) \right\|^2 \Delta t^2 + (1+\frac{1}{\alpha})C_1^2(1+\beta)\Delta t^3. \tag{32}$$

Invoking lemma 5.1, we obtain

$$\mathbb{E}[\int_{t_{k-1}}^{t_k} \left\| \mathbf{x}_s^* - \mathbf{x}_{t_{k-1}}^* \right\|^2 \mathrm{d}s] \leq 4C_0 \exp(2C_0)(C_0 + d)\Delta t^3 + 2C_0\Delta t^3 + 2d\Delta t^2.$$

Taking the expectation of eq (32), in view of the above and the assumption on control, we establish

$$\mathbb{E}[\left\| \mathbf{x}_{t_k} - \mathbf{x}_{t_k}^* \right\|^2] \leq C_3 \mathbb{E}[\left\| \mathbf{x}_{t_{k-1}} - \mathbf{x}_{t_{k-1}}^* \right\|^2] + C_4,$$

where $C_3 = [1 + \alpha + 2C_1^2(1 + \frac{1}{\alpha})(1 + \beta)(1 + \frac{1}{\eta})\Delta t^2]$, and

$$
\begin{aligned}
C_4 =& (1 + \frac{1}{\alpha})(1 + \frac{1}{\beta})d\epsilon\Delta t^2 + (1 + \frac{1}{\alpha})C_1^2(1 + \beta)\Delta t^3 \\
&+ 2C_1^2(1 + \frac{1}{\alpha})(1 + \beta)(1 + \eta)[4C_0\exp(2C_0)(C_0 + d)\Delta t^3 + 2C_0\Delta t^3 + 2d\Delta t^2]\Delta t.
\end{aligned}
$$

Finally, in view of the fact $\mathbf{x}_0 = \mathbf{x}_0^*$ and fixed step size $\Delta t$, we conclude that by the choice $\alpha = C_1\Delta t, \beta = \eta = 1$

$$
\mathbb{E}[\|\mathbf{x}_T - \mathbf{x}_T^*\|^2] \leq \frac{C_3^{\frac{T}{\Delta t}} - 1}{C_3 - 1}C_4 = \mathcal{O}(dT(\Delta t + \epsilon)), \tag{33}
$$

where the last inequality is based on ignoring the high order terms, $C_3^{\frac{T}{\Delta t}} - 1 \leq \mathcal{O}(T)$ and $\frac{C_4}{C_3 - 1} \leq \mathcal{O}(d(\Delta t + \epsilon))$.

## D    PROOF OF THEOREM 3

The proof is a natural extension of Corollary 7 in Thijssen & Kappen (2015).

We define random variable

$$
S^u(t) = \int_0^t \frac{\|\mathbf{u}_s\|^2}{2}\mathrm{d}s + \mathbf{u}_s'\mathrm{d}\mathbf{w}_s + \Psi(\mathbf{x}_T), \tag{34}
$$

and

$$
\Phi(t) = \exp(-S^u(0) + S^u(t)). \tag{35}
$$

**Lemma 5.2.** *(Thijssen & Kappen, 2015, Lemma 4) For any feasible control policy* $\mathbf{u}$ *for stochastic optimal control problem,*

$$
\Phi(T)\phi_t(\mathbf{x}_T) - \Phi(t)\phi_t(\mathbf{x}_t) = \int_t^T \Phi(s)\phi_t(\mathbf{x}_s)(\mathbf{u}_s^* - \mathbf{u}_s)'\mathrm{d}\mathbf{w}_s. \tag{36}
$$

**Corollary 1.**

$$
\phi_t(\mathbf{x}) = \mathbb{E}_{\mathcal{Q}^0}[\exp(-\Psi(\mathbf{x}_T))|\mathbf{x}_t = \mathbf{x}] = \mathbb{E}_{\mathcal{Q}^u}[\exp(-\Psi(\mathbf{x}_T) - \int_t^T \frac{\|\mathbf{u}_s\|^2}{2}\mathrm{d}s)|\mathbf{x}_t = \mathbf{x}] \tag{37}
$$

*Proof.* This follows importance sampling with density ratio from eq (11). $\qquad\square$

**Proof of Theorem 3:** We denote the important weight by $w^u$; note it is a random variable. It follows that

$$
w^u = \frac{\exp(-\Psi(\mathbf{x}_T) - \int_0^T(\frac{\|\mathbf{u}\|^2}{2}\mathrm{d}t + \mathbf{u}'\mathrm{d}\mathbf{w}))}{\mathbb{E}_{\mathcal{Q}^u}[\exp(-\Psi(\mathbf{x}_T) - \int_0^T(\frac{\|\mathbf{u}\|^2}{2}\mathrm{d}t + \mathbf{u}'\mathrm{d}\mathbf{w}))]} = \frac{\exp(-S^u(t))}{\mathbb{E}_{\mathcal{Q}^u}[\exp(-S^u(t))]}. \tag{38}
$$

Dividing the LHS of eq (36) by $\phi_0(\mathbf{x}_0)$ we obtain

$$
\frac{\Phi(T)\phi_t(\mathbf{x}_T) - \Phi(t)\phi_t(\mathbf{x}_t)}{\phi_0(\mathbf{x}_0)} = \frac{\Phi(T)\phi_t(\mathbf{x}_T) - \phi_0(\mathbf{x}_0)}{\mathbb{E}_{\mathcal{Q}^u}[\exp(-S^u(0))]} = w^u - \mathbb{E}_{\mathcal{Q}^u}[w^u]. \tag{39}
$$

Therefore, the variance of $w^u$ equals

$$
\begin{aligned}
\mathbb{E}_{\mathcal{Q}^u}[(w^u - \mathbb{E}_{\mathcal{Q}^u}[w^u])^2] &= \mathbb{E}_{\mathcal{Q}^u}[(\int_0^T \frac{\Phi(s)\phi_s(\mathbf{x}_s)}{\phi_0(\mathbf{x}_0)}(\mathbf{u}_s^* - \mathbf{u}_s)'\mathrm{d}\mathbf{w})^2] \\
&= \mathbb{E}_{\mathcal{Q}^u}[\int_0^T \frac{\Phi^2(s)\phi_s^2(\mathbf{x}_s)}{\phi_0^2(\mathbf{x}_0)}(\mathbf{u}_s^* - \mathbf{u}_s)'(\mathbf{u}_s^* - \mathbf{u}_s)\mathrm{d}s] \\
&= \mathbb{E}_{\mathcal{Q}^u}[\int_0^T (w^u\phi_s(\mathbf{x}_s)\exp(S^u(s)))^2(\mathbf{u}_s^* - \mathbf{u}_s)'(\mathbf{u}_s^* - \mathbf{u}_s)\mathrm{d}s]. \tag{40}
\end{aligned}
$$

By Jensen's inequality

$$\phi(s, \mathbf{x}_s)^2 = (\mathbb{E}_{\mathcal{Q}^u}[\exp(-S^u(s))|\mathbf{x}_s])^2 \leq \mathbb{E}_{\mathcal{Q}^u}[\exp(-2S^u(s))|\mathbf{x}_s].$$

Plugging the above inequality into eq (40), we reach the upper bound of variance

$$\mathbb{E}_{\mathcal{Q}^u}[(w^u - \mathbb{E}_{\mathcal{Q}^u}[w^u])^2] \leq \int_0^T \mathbb{E}_{\mathcal{Q}^u}[(\mathbf{u}_s^* - \mathbf{u}_s)'(\mathbf{u}_s^* - \mathbf{u}_s)(w^u)^2]\mathrm{d}s. \tag{41}$$

In view of the fact $\mathrm{Var}(w^u) + 1 = \mathbb{E}_{\mathcal{Q}^u}[(w^u)^2]$, we arrive at

$$1 + \mathbb{E}_{\mathcal{Q}^u}[(w^u)^2] \leq \mathbb{E}_{\mathcal{Q}^u}[(w^u)^2] \int_0^T \mathbb{E}_{\mathcal{Q}^u}[(\mathbf{u}_s^* - \mathbf{u}_s)'(\mathbf{u}_s^* - \mathbf{u}_s)]\mathrm{d}s.$$

If we consider the near optimal policy such that $\max_{t,\mathbf{x}} \|\mathbf{u}_t(\mathbf{x}) - \mathbf{u}_t^*(\mathbf{x})\|^2 \leq \frac{\epsilon}{T}$, then it follows that

$$\frac{1}{\mathbb{E}_{\mathcal{Q}^u}[(w^u)^2]} \geq 1 - \epsilon. \tag{42}$$

## E  PROOF OF THEOREM 4

We consider the KL divergence between trajectory distribution resulted from the policy $\mathbf{u}$ and the one from optimal policy $\mathbf{u}^*$:

$$\begin{aligned}
D_{\mathrm{KL}}(\mathcal{Q}^u(\tau)\|\mathcal{Q}^*(\tau)) &= D_{\mathrm{KL}}(\mathcal{Q}^u(\tau)\|\mathcal{Q}^0(\tau)\frac{\mu(\mathbf{x}_T)}{\mu^0(\mathbf{x}_T)}) \\
&= \mathbb{E}_{\tau \sim \mathcal{Q}^u}[\int_0^T \frac{1}{2}\|\mathbf{u}_t\|^2\,\mathrm{d}t + \mathbf{u}_t'\mathrm{d}\mathbf{w}_t + \Psi(\mathbf{x}_T)] \\
&= \mathbb{E}_{\tau \sim \mathcal{Q}^u}[\int_0^T \frac{1}{2}\|\mathbf{u}_t\|^2\,\mathrm{d}t + \mathbf{u}_t'\mathrm{d}\mathbf{w}_t + \hat{\Psi}(\mathbf{x}_T) + \log Z] \\
&= \mathbb{E}_{\tau \sim \mathcal{Q}^u}[\hat{S}^u(\tau) + \log Z] \\
&\geq 0.
\end{aligned}$$

The last inequality is based on the fact $D_{\mathrm{KL}}(\mathcal{Q}^u(\tau)\|\mathcal{Q}^*(\tau)) \geq 0$ and the equality holds only when $\mathbf{u} = \mathbf{u}^*$, pointing to

$$0 = \mathbb{E}_{\tau \sim \mathcal{Q}^*}[\hat{S}^u(\tau) + \log Z].$$

Therefore, we can estimate the normalization constant by

$$Z \geq \exp(-\mathbb{E}_{\tau \sim \mathcal{Q}^u}[\hat{S}^u(\tau)]), \quad Z = \exp(-\mathbb{E}_{\tau \sim \mathcal{Q}^*}[\hat{S}^u(\tau)]). \tag{43}$$

Next we provide an unbiased estimation with sub-optimal policy $\mathbf{u}$ based on importance sampling as

$$\begin{aligned}
1 &= \mathbb{E}_{\tau \sim \mathcal{Q}^u}[\frac{\mathcal{Q}^*(\tau)}{\mathcal{Q}^u(\tau)}] \\
&= \mathbb{E}_{\tau \sim \mathcal{Q}^u}[\exp(\log \frac{\mathcal{Q}^*(\tau)}{\mathcal{Q}^u(\tau)})] \\
&= \mathbb{E}_{\tau \sim \mathcal{Q}^u}[\exp(-\hat{S}^u(\tau) - \log Z)].
\end{aligned}$$

The last equality is based on the fact $\mathbb{E}_{\tau \sim \mathcal{Q}^u}[\frac{\mathcal{Q}^*(\tau)}{\mathcal{Q}^u(\tau)}] = \int_\tau \mathcal{Q}^*(\tau)\mathrm{d}\tau = 1$. Hence, we obtain an unbiased estimation of the normalization constant as

$$Z = \mathbb{E}_{\tau \sim \mathcal{Q}^u}[\exp(-\hat{S}^u(\tau))].$$

## F  EXPERIMENT DETAILS AND DISCUSSIONS

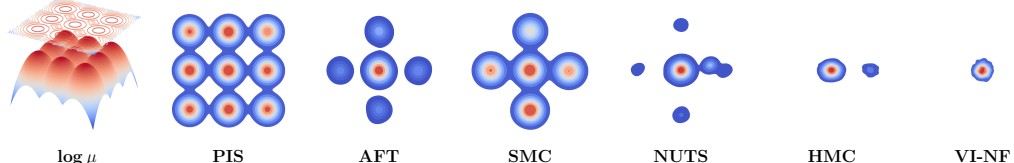

Figure 4: Sampling performance on a challenging 2D unnormalized density model with well-separated modes. Kernel density estimation plots are compared with 2k samples. AFT and SMC use annealing trick with 10 decreasing temperate levels and HMC kernel following (Arbel et al., 2021). Even without annealing trick and resampling, Path Integral Sampler (PIS) generates visually indistinguishable samples from target density with 100 steps. PIS starts $\mathbf{x}_0$ from origin point while others start from a standard Gaussian. The underlying distribution is chosen deliberately to distinguish the performance of different methods. In particular, 100 steps are not sufficient for general MCMC to converge to the stationary distribution. We also note performance of compared methods can be further improved with tuning temperature scheduling, samples initialization. Our generic algorithm can explore more modes with similar initialization and less tuning parameters.

### F.1    TRAINING TIME, MEMORY REQUIREMENTS AND SAMPLING EFFICIENCY

The PIS can be trained once before sampling and did not contribute to the runtime in sampling computation. Most MCMC methods do not have learnable parameters. However, PIS policy is trained once and used everywhere. Thus, the training time can be amortized when PIS policy is deployed in generating a large number of samples. The training time of PIS highly depends on efficiency of training NeuralSDEs. One future direction is to investigate additional regularizations and structured SDEs to speed up the training. The PIS algorithm is implemented in PyTorch (Paszke et al., 2019). We use Adam optimizer (Kingma & Ba, 2014) in all experiments to learn optimal policy with learning rates $5 \times 10^{-3}$ and other default hyperparameters. All experiments are trained with 30 epochs and 15000 points datasets. Loss in most experiments plateau after 3 epochs, some even 1 epoch. Experiments are conducted using an NVIDIA A6000 GPU. Training one epoch on 2d example takes around 15 seconds for PIS-NN and 30 seconds for PIS-Grad, 1.6 minutes and 1.8 minutes respectively on Funnel ($d = 10$), and 7 minutes and 9 minutes on LGCP ($d = 1600$). We note both PIS and AFT can be trained once and used everywhere. Therefore, the training time can be amortized when the PIS policy is deployed in generating a large number of samples. We further compare empirical sampling time for various approaches in Tab 4.

In the high dimensional data, thanks to the efficient adjoint SDE solver, we do not need to cache the whole computational graph and the required memory is approximately the cost associated with one forward and backward pass of $\mathbf{u}_t(\mathbf{x})$ network. In our experiments, the total consumed memory is around 1.5GB for the toy and Funnel example, and around 5GB for the LGCP.

| method | Sampling Time | Training Time |
|---|---|---|
| AFT | 352.2ms | 711.2ms |
| SMC | 110.8 ms | – |
| PIS-NN | 16.8 ms | 30.3 ms |
| PIS-Grad | 34.3 ms | 61.2 ms |
| SNF | 130.6 ms | 256.1 ms |

Table 4: Sampling and training efficiency comparison. For sampling, we measure the consumed time for generating 2000 particles with each method for 100 times in lower dimensional data, 2-D points datasets, and report average time for each method. We also include one batch training time for AFT and PIS.

### F.2    NETWORK INITIALIZATION

In most experiments in the paper, we found PIS with the default setting, $T = 1$ and zero control initialization, gives reasonable performance. One exception is LGCP, where we found training with $T = 1$ sometimes suffers from numerical issues and gives NAN loss. We sweep $T = 1, 2, 5$ and found $T = 5$ gives encouraging results. As we discussed in appendix G, the optimal policy $\mathbf{u}^*$ depends on $T$ and large $T$ not only results in a large cover area but also a "smoother" $\mathbf{u}^*$. For neural network weight initialization, we use the zero initialization for the last layer of parameterized policy $\mathbf{u}_t(\mathbf{x})$ and other weights follow default initialization in PyTorch. We note there is no guarantee that PIS can cover all modes initially with only the given unnormalized distribution. However, the initialization problem exists for most sampling algorithms and MCMC algorithms suffer longer mixing times compared with the ones with proper initialization (Chopin & Papaspiliopoulos, 2020). PIS also suffers from improper initialization and we report some failure mode in appendix G.

### F.3 DETAILS FOR COMPARED METHODS

For all trained PIS and its variants, we use uniform 100 time-discretization steps for the SDEs. Gradient clipping with value 1 is used. A Fourier feature augmentation (Tancik et al., 2020) is employed for time condition. Throughout all experiments, we use the same network with the modified first layer and last layer for different input data shape. In one pass $\mathbf{u}_t(\mathbf{x})$, we augmented scalar $t$ with Fourier feature to 128 dimension, followed by a 2 linear layers to extract 64 dimension signal feature. We use 2 layer linear layers to extract 64 dimension feature for $x$ and the concatenate $x, t$ features before feeding into another 3 linear layer with 64 hidden neurons to get $\mathbf{u}_t(\mathbf{x})$. We found that the training is stable with our simple MLP parametrization in our experiments.

For HMC, we use 10 iterations of Hamiltonian Monte Carlo with 10 leapfrog steps per iterations, totaling 100 leapfrog steps. For NUTS, we set the maximum depth of the tree built as 5. Note that samples of HMC and NUTS used in our experiments are from separate trajectories instead of from one trajectory at different timestamps. We observed that the latter is more likely to generate samples that concentrate on one single mode. For SMC and AFT, we use 10 transitions with each transition using the same amount computation as HMC. The settings of SMC and AFT follow the official implementation (Arbel et al., 2021) in the released codebase [1]. In the Alanine Dipeptide experiments, for AFT we sweep adam learning rate $lr = 1 \times 10^{-4}, 5 \times 10^{-4}, 1 \times 10^{-3}, 5 \times 10^{-3}, 1 \times 10^{-2}$ and select $5 \times 10^{-3}$. Other settings follow the default setup from the official codebase.

### F.4 COMPARISON WITH SVGD ALGORITHM

In this section, we present some comparison between celebrated SVGD sampling algorithm (Liu et al., 2016; 2019) and PIS sampler. SVGD is based on collections of interacting particle, and we found its performance increase with more samples in a batch as it shown in Fig 5. Even with 5000 particles in a bach, sample quality of SVGD is still worse than PIS-Grad. Meanwhile, the kernel-based approach pays much more computation results as the number of active particles increases as we show in Tab 5 compared with PIS. In this perspective, PIS enjoy much better scalibility.

| method | Time (mean $\pm$ std) |
|---|---|
| SVGD 100 particles | 19.3 ms $\pm$ 1.2 ms |
| SVGD 1000 particles | 29.4 ms $\pm$ 1.8 ms |
| SVGD 5000 particles | 434 ms $\pm$ 2.43 ms |
| PIS-NN 5000 particles | 223 μs $\pm$ 14.2 μs |
| PIS-Grad 5000 particles | 412 μs $\pm$ 15.1 μs |

Table 5: Consumed time for one iteration of SVGD and one step in PIS in 2D points example. Sampling with SVGD is less efficient compared with PIS.

---

[1]https://github.com/deepmind/annealed_flow_transport

| 100 particles | 1000 particles | 5000 particles |

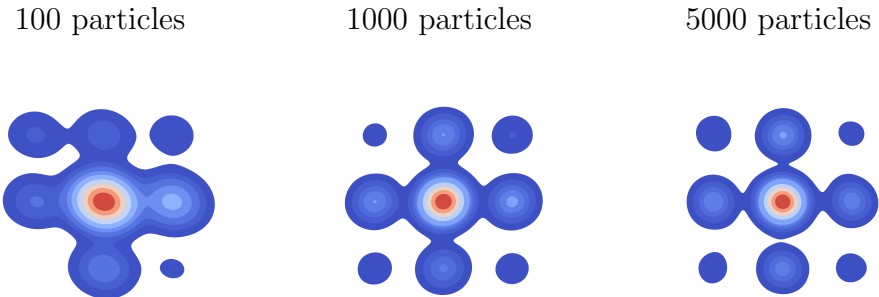

Figure 5: Generated samples from SVGD (Liu & Wang, 2016) with 100 steps. We generated samples with batch size 100, 1000, 5000. We find with more particles, samples generated are more closed to the ground truth data.

### F.5 CHOICE OF PRIOR SDE:

We can use a more general SDE

$$d\mathbf{x}_t = \mathbf{f}(t, \mathbf{x}_t)dt + \mathbf{g}(t, \mathbf{x}_t)(\mathbf{u}_t dt + d\mathbf{w}_t), \ \ \mathbf{x}_0 \sim \nu \qquad (44)$$

instead of eq (2) for sampling. As discuss in Section 3.3, we prefer use a linear function for $\mathbf{f}, \mathbf{g}$ to promise a closed-form $\mu^0$. The choice of $\mathbf{f}, \mathbf{g}$ encodes prior knowledge into dynamics without control and $\mathcal{Q}^*$ is determined based on the prior $\mathcal{Q}^0$. Intuitively, the ideal $\mathcal{Q}^0$ should drive particles from $\nu$ to $\mathcal{Q}^0(\mathbf{x}_T)$ that is close to $\mu$. In PIS, our training objective is to fit $\mathcal{Q}^*$ with parameterized $\mathcal{Q}^u$. Thus training can be easier and faster if $\mathcal{Q}^*$ and $\mathcal{Q}^0$ are close since we use zero control as initialization for training policy. However, there is no general approach to choose $\mathbf{f}, \mathbf{g}$ such that $\mathcal{Q}^0(\mathbf{x}_T)$ is close to $\mu$ and $\mathcal{Q}^0(\mathbf{x}_T)$ has a closed form. In this work, we adopt the general form with $\mathbf{f} = 0, \mathbf{g} = \mathbf{I}$. It would be interesting to explore other prior dynamics or data-variant $\mathbf{f}, \mathbf{g}$ in the future work, e.g., underdamped Langevin.

### F.6 ESTIMATION OF NORMALIZATION CONSTANTS

As discussed in Chopin & Papaspiliopoulos (2020), normalization constants estimation of SMC and its variants AFT can be achieved with incremental importance sampling weights.

In our experiments we treat HMC and NUTS as special cases of SMC with only two different temperature levels. One corresponds to a standard Gaussian distribution and the other one corresponds to the target density. Since the initial distribution $\nu$ for SMC and NUTs is chosen as standard Gaussian, we can omit the MCMC steps for it and the total computation efforts required for the specific SMC are for the transitions in HMC and NUTS.

For VI-NF, we use importance sampling

$$\int \hat{\mu}(\mathbf{x})d\mathbf{x} = \int q(\mathbf{x})\frac{\hat{\mu}(\mathbf{x})}{q(\mathbf{x})}d\mathbf{x} = \mathbb{E}_q[\frac{\hat{\mu}(\mathbf{x})}{q(\mathbf{x})}],$$

where $q$ is the normalized distribution represented by normalizing flows, to provide an unbiased estimation of normalization constants. We use the ELBO in eq (18) for PIS and the unbiased estimation eq (19) for $\text{PIS}_{RW}$.

### F.7 2 DIMENSIONAL RINGS EXAMPLE

The ring-shape density function

$$\log \hat{\mu} = -\frac{\min((\|\mathbf{x}\| - 1)^2, (\|\mathbf{x}\| - 3)^2, (\|\mathbf{x}\| - 5)^2)}{100}.$$

Consider the special case of gradient informed SDE, which can be viewed as PIS-Grad with a specific group of parameters,

$$\mathrm{d}\mathbf{x}_t = \nabla \log \hat{\mu}(\mathbf{x}_t)\mathrm{d}t + \sqrt{2}\mathrm{d}\mathbf{w}_t.$$

This is exactly the Langevin dynamics used widely in sampling (MacKay, 2003). As a special case of MCMC, Langevin sampling can generate high quality samples given large enough time interval (MacKay, 2003). From this perspective, PIS-Grad can be viewed as a modulated Langevin dynamics that is adjusted and represented by neural networks.

## F.8    BENCHMARKING DATASETS

| | MG(d=2) | | | Funnel(d=10) | | | LGCP(d=1600) | | |
| | B | S | $A$ | B | S | $A$ | B | S | $A$ |
|---|---|---|---|---|---|---|---|---|---|
| AFT-$10^3$ | -0.509 | 0.24 | 0.562 | -0.249 | 0.0758 | 0.261 | -3.08 | 1.59 | 3.46 |
| SMC-$10^3$ | -0.362 | 0.293 | 0.466 | -0.338 | 0.136 | 0.364 | -440 | 14.7 | 441 |
| AFT-$2 \times 10^3$ | -0.371 | 0.477 | 0.604 | -0.249 | 0.0758 | 0.261 | -1.23 | 0.826 | 1.48 |
| SMC-$2 \times 10^3$ | -0.398 | 0.198 | 0.444 | -0.338 | 0.136 | 0.364 | -197 | 5.21 | 197 |
| AFT-$3 \times 10^3$ | -0.316 | 0.365 | 0.483 | -0.281 | 0.0839 | 0.293 | -1.05 | 0.514 | 1.17 |
| SMC-$3 \times 10^3$ | -0.137 | 0.62 | 0.635 | -0.323 | 0.064 | 0.329 | -109 | 5.58 | 109 |
| AFT-$5 \times 10^3$ | -0.194 | 0.319 | 0.373 | -0.253 | 0.0397 | 0.256 | -0.949 | 0.439 | 1.05 |
| SMC-$5 \times 10^3$ | -0.129 | 0.246 | 0.278 | -0.298 | 0.0564 | 0.303 | -37.5 | 5.04 | 37.8 |
| AFT-$10^4$ | -0.03 | 0.515 | 0.515 | -0.194 | 0.0554 | 0.202 | **-0.827** | **0.356** | **0.901** |
| SMC-$10^4$ | -0.171 | 0.446 | 0.477 | -0.239 | 0.0412 | 0.243 | -6.47 | 1.95 | 6.76 |
| PIS-$10^2$ | **-0.021** | **0.03** | **0.037** | **-0.008** | **9e-3** | **0.012** | -1.94 | 0.91 | 2.14 |

Table 6: Long-run MCMC on mode separated mixture of Gaussian (MG), Funel distribution and Log Gaussian Cox Process (LGCP) for estimating log normalization constants. The suffix denotes the total number of discrete-time steps for each method, which equals the number of layers multiply steps per layer. We experiments $10, 20, 30, 50, 100$ layers for annealing and $100$ leapfrog steps per layer. As the number of steps increases, the performance of AFT and SMC gradually improves. PIS denotes the PIS$_{RW}$-Grad. $B$ and $S$ stand for estimation bias and standard deviation among 100 runs and $A^2 = B^2 + S^2$.

For mixture of Gaussian, we choose nine centers over the grid $\{-5, 0, 5\} \times \{-5, 0, 5\}$, and each Gaussian has variance $0.3$. The small variance is selected deliberately to distinguish the performance of the different methods. We use 2000 samples for estimating the log normalization constant $Z$. We use the standard MLP network to parameterize the control drift $\mathbf{u}_t(\mathbf{x})$, where the time signal is augmented by Fourier feature using 64 different frequencies. We use 2 layer (64 hidden neurons in each layer) MLP to extract features from the augmented time signal and $\mathbf{x}$ separately, and another 2 layer MLP to map the summation of features to the policy command. We note that all these methods for comparison, including HMC, NUTS, SMC, AFT, can reach reasonably good results given large enough iterations. However, with small finite number of steps, PIS achieves the best performance. We include more results for long-run MCMC methods in Tab 6.

In the experiment with Funnel distribution, Arbel et al. (2021) suggests to use a slice sampler kernel for AFT and SMC, which includes 1000 steps of slice sampling per temperature. In Tab 1, we still use HMC for comparing performance with the same number of integral steps. We also include the results with slice sampler in Tab 7. We use 6000 particles for the estimation of log normalization constants. The network architecture of PIS is exactly the same as that in the experiments with mixture of Gaussian.

In the example with Cox process, the covariance $K$ is chosen as

$$K(u, v) = 1.91 \times \exp(-\frac{\|u - v\|}{M\beta}),$$

|  | Funnel(d=10) | | |
|---|---|---|---|
|  | B | S | $A$ |
| AFT-10 | 0.128 | 0.376 | 0.398 |
| SMC-10 | -0.193 | 0.067 | 0.204 |
| AFT-20 | 0.0134 | 0.173 | 0.174 |
| SMC-20 | -0.113 | 0.0878 | 0.143 |
| AFT-30 | 0.074 | 0.309 | 0.318 |
| SMC-30 | **-0.006** | 0.188 | 0.188 |
| $PIS_{RW}$-Grad | -0.008 | **0.009** | **0.012** |

Table 7: AFT and SMC with slice sampler kernel. The suffix denotes the number of temperature levels for annealing. 1000 slicing sampling steps are used for each temperature. Though there is no annealing and only 100 steps are used, the performance of PIS is competitive.

and the mean vector equals $\log(126) - \sigma^2$ and $\alpha = 1/M^2$. We note that this setting follows Arbel et al. (2021); Møller et al. (1998). Totally 2000 samples are used to evaluate the log normalization constant. We treat the mean of estimation results from 100 repetitions of SMC with 1000 temperatures as ground truth normalization constants. In this experiment, we found clipping gradient from target density function help stabilize and speed up the training of PIS-Grad. This example is the most challenging task among the three. One major reason is the high dimensionality of the task; the PIS needs to find optimal policy $\mathbf{u} : (t, \ \mathbb{R}^d) \to \mathbb{R}^d$ in high dimensional space. In addition, there is no prior information that can be used to shrink the search space, which makes the training of PIS with MLP more difficult. We use 2000 particles for estimation of log normalization constants. We also include more experiment results in Tab 8.

|  | LGCP(d=1600) | | |
|---|---|---|---|
|  | B | S | $A$ |
| AFT-$10^4$ | **-0.827** | 0.356 | 0.901 |
| $PIS_{RW}$-Grad-$1 \times 10^2$ | -1.94 | 0.91 | 2.14 |
| $PIS_{RW}$-Grad-$5 \times 10^2$ | -1.25 | 0.57 | 1.373 |
| $PIS_{RW}$-Grad-$10 \times 10^2$ | -0.832 | **0.214** | **0.859** |

Table 8: PIS with large number of integral step. The suffix number is the total integral steps. For AFT-$10^4$, we use 100 annealing layers and run 100 leapfrog steps per each annealing layer.

### F.9   ALANINE DIPEPTIDE

The setup for target density distribution and the comparison method are adopted from Wu et al. (2020). Following Noé et al. (2019), an invertible transformation between Cartesian coordinates and the internal coordinates is deployed before output the final samples. Then we normalize the coordinates by removing means and dividing them by the standard deviation of train data. To setup the target distribution, we simulate Alanine dipeptide in vacuum using OpenMMTools (Eastman et al., 2017) [2]. Total $10^5$ atoms data points are generated as training data. Situation parameters, including time-step and temperature setting are the same as Wu et al. (2020). We refer the reader to official codebase for more details of setting target density function [3].

Following the setup in Wu et al. (2020), we use unweighted samples to compute the metrics. KL divergence of VI-NF on $\mu$ is calculated based on ELBO instead of importance sampling as in normalizing constants tasks. We use Metropolis random walk MCMC block for SMC, SNF and AFT and RealNVP blocks for SNF and AFT (Dinh et al., 2016). For a fair comparison, we use PIS-NN instead of PIS-Grad since none of the approaches in this example uses the gradient information. We note that SNF is originally trained with maximizing data likelihood where some empirical samples

---
[2] https://github.com/choderalab/openmmtools
[3] https://github.com/noegroup/stochastic_normalizing_flows

are assumed to be available. We modify the training objective function by reversing the original KL divergence as in eq (1).

### F.10 MORE DETAILS ON SAMPLING IN VARIATIONAL AUTOENCODER LATENT SPACE

Figure 6: Origin data images and their reconstructions from trained vanilla VAE. It can be seen that reconstruction images are smoother compared with the original images.

We use a vanilla VAE architecture to train on binary MNIST data. The encoder uses a standard 3 layer MLP networks with 1024 hidden neurons, and maps an image to the mean and standard deviation of 50 dimension diagonal Normal distribution. The decoder employs 3 layer MLP networks to decode images from latent states. ReLU nonlinearity is used for hidden layers. For training, we use the Adam optimizer with learning rate $5 \times 10^{-4}$, batch size 128. With reparameterization trick (Kingma & Welling, 2013) and closed-form KL divergence between approximated normal distribution and standard normalization distribution, we train networks for totally 100 epochs. We show performance of vanilla in Fig 6.

We parameterize distribution of decoder $p_\theta(\mathbf{x}|\mathbf{z})$ as

$$\log p_\theta(\mathbf{x}|\mathbf{z}) = \log p(\mathbf{x}|D_\theta(\mathbf{z})) = \mathbf{x} \log D_\theta(\mathbf{z}) + (1 - \mathbf{x}) \log(1 - D_\theta(\mathbf{z})).$$

For PIS, we use the same network and training protocol as that in the experiment for mixture of Gaussian and Funnel distributions. We also use gradient clip to prevent the magnitude of control drift from being too large.

## G TECHNICAL DETAILS AND FAILURE MODES

### G.1 TIPS

Here we provide a list of observations and failure cases we encountered when we trained PIS. We found such evidences through some experiments, though there is no way we are certain the following claims are correct and general for different target densities.

- We notice that smaller $T$ may result in control strategy with large Lipchastiz constants, which is within expectation since large control is required to drive particles to destination with less amount of time. It is reported that it is more difficult to approximate large Lipchastiz functions with neural networks (Jacot et al., 2018; Tancik et al., 2020). We thus recommend to increase $T$ or constraint the magnitude of $\mathbf{u}$ to stablize training when encountering numeric issue or when results are not satisfactory.

- We found batch normalization can help stablize and speed up training, and the choice of nonlinear activation (ReLU and its variants) does not make much difference.

- We also notice that if the control $\mathbf{u}_t(\mathbf{x})$ has large Lipchastiz constants in time dimension, the discretized error would also increase. For calculating the weights based on path integral, we suggest to decrease time stepsize and increase $N$ when the number of integral steps is small and discretization error is high.

- We obtained more stable and smaller training loss when training with Tweedie's formula (Efron, 2011), but we found no obvious improvements on testing the trained sampler or estimating normalization constants.

- Regardless the accuracy and memory advantages of Reversible Heun claimed by *torchsde* (Li et al., 2020a; Kidger et al., 2021) , we found this integration approach is

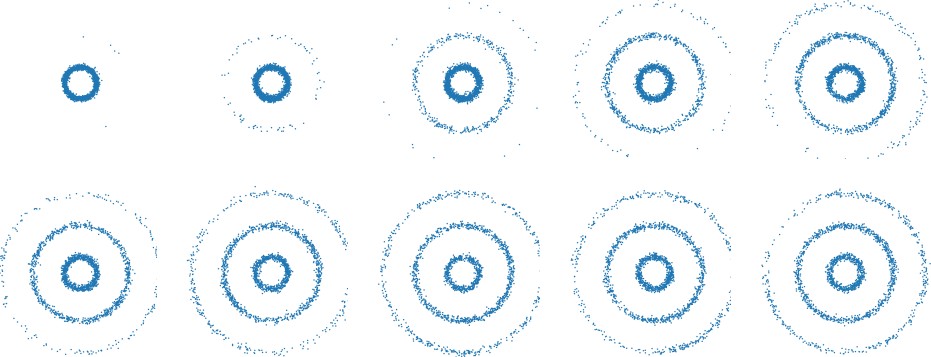

Figure 7: Generated 5000 uncurated samples with $T = 0.1, 0.2, 0.4, 0.6, 0.8, 1.0, 2.0, 3.0, 4.0, 5.0$. PIS with small T may miss some modes.

less stable compared with simple Euler integration without adjoint and results in numerical issues occasionally. We empirically found that methods without adjoint are more stable and lower loss compared with adjoint ones, even in low dimensional data like 2d points. We use Euler-Maruyama without adjoint for low dimension data and Reversible Heun method (Kidger et al., 2021) in datasets when required memory is overwhelming to cache the whole computational graph. We recommend readers to use Ito Euler integration when memory permits or conduct training with a small $\Delta t$. The more steps in PIS and smaller $\Delta t$ not only reduce policy discretization error Theorem 2, but also reduce the Euler-Maruyama SDE integration errors.

### G.2 FAILURE MODES

In this section, we show some failure modes of PIS. With an improper initialization and an extremely small $T$, PIS suffers from missing mode. The failure can be resulted from following factors. First, the untrained and initial $p(\mathbf{x}_T) = N(0, T\mathbf{I})$ may be far away from the target the distribution modes. Thus it is extremely difficult for training PIS to cover region of high probability mass under $\mu$. Second, small $T$ results in policies with large Lipchastic constants that are challenging to train networks as we discussed in appendix G.1. Third, the policy with limited representation power tends to miss modes due to the proposed KL training scheme. We show some uncurated samples trained with different $T$ and other failure cases.

Thoughout our experiments, we also find that large $T$ leads to more stable training and PIS sampler covers more modes. However large $T$ with large $\delta t$ deteriorates sample quality due to discretization error but it can be eased with increasing number of steps.

We note PIS is not a perfect sampler and failure modes exists for all compared methods, experiments results in Section 4 we reported are based on uncurated experiments.

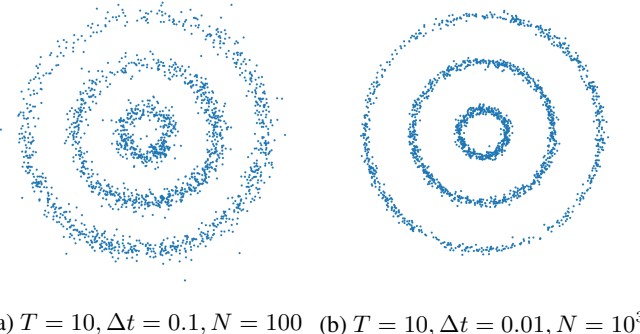

(a) $T = 10, \Delta t = 0.1, N = 100$   (b) $T = 10, \Delta t = 0.01, N = 10^3$

Figure 8: Large $T$ with large $\delta t$ deteriorates sample quality due to discretization error in Fig 8a but it can be eased with increasing number of steps in Fig 8b.

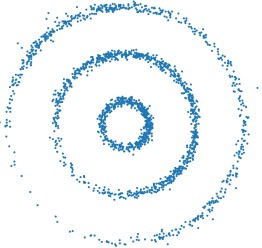

Figure 9: Another failure case with $T = 2, \Delta t = 0.02, N = 100$ due to randomness of training networks.

