# OpenReview forum: "Path Integral Sampler: A Stochastic Control Approach For Sampling"
_ICLR.cc/2022/Conference — ICLR 2022 Poster_

### Official Review · Reviewer_ZcgD · 2021-10-31

**Correctness:** 4
**Technical Novelty And Significance:** 3
**Empirical Novelty And Significance:** 3
**Recommendation:** 8
**Confidence:** 4

**Main Review:**

Strengths: clear exposition, theoretical guarantees provided, informative experiments and ablations

Weaknesses:
0) It would be informative to see a comparison to Stein-Variational Gradient Descent (SVGD) or other ParVI methods [1]
1) The paper says "To calculate gradients, we rely on backpropagation through trajectory." Are there any issues associated with that? E.g., exploding/vanishing gradient. How long can trajectories be? How much memory/time is required?
2) In Algorithm 1, specify what is input and output of the algorithm
3) In Algorithm 2, it is a bit confusing that 'w' is used as the argument of the exponential, which is different from Eq. (17), where it is used as the exponential itself
4) In Theorem 2, Condition 1 is only defined in the Appendix. Either state the condition before the theorem or expand it in the statement of the theorem
5) The paper says "training consumption is not included in the comparison". Could the authors state at least the order of magnitude of how much time the proposed method takes compared to the baselines?
6) In Table 1, highlighting PIS_RW-GT may be seen as a distraction, since it is using the ground truth. Therefore, perhaps one could either separate that line or set those numbers in italic and use bold on PIS_RW-Grad for MG(d=2) columns
7) "more clearer" -> "clearer"; "as it in a toy example" -> "as in a toy example"
8) What are the limitations? Adding a paragraph on drawbacks and limitations would be helpful.

[1] Liu, C., Zhuo, J., Cheng, P., Zhang, R., & Zhu, J. (2019, May). Understanding and accelerating particle-based variational inference. In International Conference on Machine Learning (pp. 4082-4092). PMLR.

**Summary Of The Paper:**

The paper proposes an algorithm called Path Integral Sampler (PIS) for sampling from unnormalized distributions by parameterizing the control policy in the Schrödinger bridge problem with neural networks and supplying it with the gradient information from the target density. Performance guarantees are provided, and experiments on several illustrative toy problems as well as on sampling molecular structures and latent space of VAEs are conducted. The method is shown to produce high-quality samples from the posterior, demonstrating state-of-the-art performance.

**Summary Of The Review:**

The paper is overall good. The method is based on taking the established framework of inference as control via Schrödinger bridge problem and utilizing NNs as a parameterization for the policy. Exposition is clear, convergence is characterized, the properties of the algorithm are highlighted on relatively low-dimensional problems.

====

After rebuttal

I thank the authors for addressing my comments. I trust that the authors will include the promised changes and additions regarding the training time and the comparison to ParVI into the final version of the paper. I maintain my score "8: accept, good paper".

---

> ### Author Response · Authors · 2021-11-18
> **Response for Reviewer ZcgD**
>
> **It would be informative to see a comparison to Stein-Variational Gradient Descent (SVGD) or other ParVI methods [1]**
>
> The suggested works can generate samples from a given target distribution. However, this type of method differs significantly from MCMC; they rely on a collection of interacting particles and are not guaranteed to generate unbiased samples (though they often have good performance, especially when the number of samples is small).
>
>
> We will include a comparison between SVGD and PIS in the revision. The [link](https://anonymousau.github.io/uuttqpage/letter/svgd) shows some experimental results we obtained. In this simple example, PIS-Grad is able to generate better samples (according to a human observer) and more efficient.
>
> **The paper says "To calculate gradients, we rely on backpropagation through trajectory." Are there any issues associated with that? E.g., exploding/vanishing gradient. How long can trajectories be? How much memory/time is required?**
>
> We have a short discussion on the Neural SDE solver in Appendix G (technical tips). We use [torchsde](https://github.com/google-research/torchsde/commit/53038a3efcd77f6c9f3cfd0310700a59be5d5d2d)[3] in our implementation, specifically, we use Euler-Maruyama without adjoint for low dimension data and Reversible Heun method in datasets when required memory is overwhelming to cache the whole computational graph. We empirically found that methods without adjoint are more stable and lower loss compared with adjoint ones, even in low dimensional data like 2d points.
>
>
> We observed numerical instability when $T$ is small. This is probably due to the fact that the magnitude and Lipchastiz constants on $t$ of optimal policy function increase when $T$ decreases. We recommend increasing $T$ when performance is not satisfying or numerical instability occurs.
> In our experiments, we did not encounter any exploding/vanishing gradient during our experiments. As is shown in [4,5], backpropagation in SDE is different from RNNs training, and the gradient is another SDE associated with adjoint states.
>
> As discussed in Appendix F, all experiments are trained with $30$ epochs and $15000$ points datasets. The loss in most experiments plateaus after 3 epochs, sometimes even after 1 epoch. All experiments are conducted using a single NVIDIA A6000 GPU. Training one epoch on 2d example takes about 15 seconds for PIS-NN and 30 seconds for PIS-Grad, 1.6 minutes and 1.8 minutes respectively on Funnel (d=10), and 7 minutes and 9 minutes on LGCP (d=1600). Thanks to the efficient adjoint SDE solver, we do not need to cache the whole computational graph and the required memory is roughly about the cost associated with one forward and backward pass of $u_t(x)$ network.
> In our experiments, the total consumed memory is around 1.5GB for the toy and Funnel example, and around 5GB for the LGCP.
>
> **In Algorithm 1, specify what is input and output of the algorithm**
>
> Algorithm 1 illustrates how to train the control policy. The required input is the $\mu$.
> The training data are the trajectories start with $x_0=0, y_0=0$ and are driven by SDE parameterized by $f_{aug}, g_{aug}$. We optimize the policy by minimizing the loss in Equation (13). The output of the algorithm is a learned policy for the given target distribution $\mu$. We will clarify this in the revision.
>
> **In Algorithm 2, it is a bit confusing that 'w' is used as the argument of the exponential, which is different from Eq. (17), where it is used as the exponential itself**
>
> We thank the reviewer for pointing it out. We will fix the notations in the revision by using a new notation in Algorithm 2.
>
> **In Theorem 2, Condition 1 is only defined in the Appendix. Either state the condition before the theorem or expand it in the statement of the theorem**
>
> Yes, we agree. We will place an informal theorem in the main paper and a detailed theorem in Appendix.

---

> > ### Author Response · Authors · 2021-11-18
> > **More Response for Reviewer ZcgD**
> >
> >
> > **The paper says "training consumption is not included in the comparison". Could the authors state at least the order of magnitude of how much time the proposed method takes compared to the baselines?**
> >
> > The PIS can be trained once before sampling and did not contribute to the runtime in sampling computation. Most MCMC methods do not have learnable parameters, so we do not include that part for comparison. However, PIS policy is trained once and used everywhere. Thus, the training time can be amortized when PIS policy is deployed in generating a large number of samples.
> >
> > The concrete training time for PIS is discussed in the answer to a previous question. For AFT, generating samples requires retraining samples. We show the empirical consumed time with generating 5000 particles in lowe dimensional data. For SNF with default settings, I find time for forward passing and backward passing is a bit larger than PIS and PIS converges much faster than SNF.
> >
> >
> > | method      | Time (mean ± std) |
> > | ------ | ----------- |
> > | AFT        | 8924.2ms ± 123.17 ms        |
> > | SMC        | 130.8 ms ± 3.21 ms        |
> > | PIS-NN     | 44.8 ms ± 1.42 ms        |
> > | PIS-Grad   | 61.3 ms ± 1.51 ms        |
> > |SNF          |  100.6 ms ± 2.13ms  |
> >
> >
> > **In Table 1, highlighting PIS_RW-GT may be seen as a distraction since it is using the ground truth. Therefore, perhaps one could either separate that line or set those numbers in italic and use bold on PIS_RW-Grad for MG(d=2) columns**
> >
> > Thanks for the suggestion. We agree with the reviewer and will use italic for the results associated with PIS_RW-GT.
> >
> > **Typos**
> >
> > Thanks again for the reviewer's careful reading. We will correct the typos.
> >
> > **What are the limitations? Adding a paragraph on drawbacks and limitations would be helpful.**
> >
> > We will add some discussions on limitations and failure modes cases. One key hyperparameter of PIS is the interval length $T$. In our implementation of parameterizing policy, we set $u_t(x)=0$ for first iteration so $X_T \sim N(0,T I)$ for the first iteration. The KL training loss function is known to have zero-forcing and may miss modes if initialization is not proper.
> >
> >
> > [1]. Liu, C., Zhuo, J., Cheng, P., Zhang, R., & Zhu, J. (2019, May). Understanding and accelerating particle-based variational inference.
> >
> > [2]. Qiang Liu, Jason Lee, Michael Jordan. A Kernelized Stein Discrepancy for Goodness-of-fit Tests.
> >
> > [3]. Patrick Kidger, James Foster, Xuechen Li, Terry Lyons. Efficient and Accurate Gradients for Neural SDEs.
> >
> > [4]. Ricky T. Q. Chen, Yulia Rubanova, Jesse Bettencourt, David Duvenaud. Neural Ordinary Differential Equations
> >
> > [5]. Xuechen Li, Ting-Kam Leonard Wong, Ricky T. Q. Chen, David Duvenaud. Scalable Gradients for Stochastic Differential Equations

---

> ### Author Response · Authors · 2021-11-30
> **revision**
>
> We thank the reviewer for the feedback.
>
> I apologize that I didn’t prepare the revision. This is my first ICLR submission and I was not aware of the possibility of uploading a revision. Thanks to reviewer 1wV8 comments, I have prepared a revision today following my response above that addresses most of the technical issues. Since uploading a revision is no longer is an option, I would like to share a revision through [this anonymous link](https://anonymousau.github.io/uuttqpage/assets/pdf/revision1.pdf).  We are considering more ParVI algorithms and plan to add them in the main paper if they have a reasonable performance in high-dimensional datasets.
>
> I sincerely hope you can take this into account in your review. We are happy to provide any further clarification and discussion.

---

### Official Review · Reviewer_dEAt · 2021-11-03

**Correctness:** 4
**Technical Novelty And Significance:** 3
**Empirical Novelty And Significance:** 3
**Recommendation:** 8
**Confidence:** 3

**Main Review:**

This paper is excellently written and great care is taken
to highlight exactly what properties of each formalism are
used to get the nice closed-form equations. The work is
highly novel and one that would be greatly valued by the
community. The proofs are clear and easy to follow as well.

The experiments are rigorous and persuasive. One thing I
don't understand is why was the work not compared against
neural density estimators? There is a rich literature of
flow models that is referenced in the paper but aside from
SNF not really compared against. Some of it even uses
neural networks that backpropagate through a SDE solver!


**Summary Of The Paper:**

This paper shows that sampling can be treated as a stochastic
optimal control program and introduces a novel sampling algorithm
that works by simulating a stochastic differential equation under
optimal control with appropriately chosen dynamics and cost function.


**Summary Of The Review:**

This work ntroduces a novel sampling algorithm with lots of interesting
properties. This paper makes a solid contribution with the strengths and
limitations of the approach very clearly articulated.

---

> ### Author Response · Authors · 2021-11-18
> **Response for Reviewer dEAt: comparison with neural density estimators**
>
> We are so glad that the reviewer appreciates this work.
>
> **Q: Missing comparison against neural density estimators**
>
> The task (density sampling) we investigate is different from the tasks of existing neural density estimators (density learning). In this work, PIS is learned to sample from a given unnormalized density function (whose analytic formula is available) without unbiased samples from the distribution. In contrast, most neural density estimators focus on learning unknown density with given unbiased data from the underlying distribution. For neural density estimators in density learning, the popular objective is to maximize the loglikelihood (a.k.a. minimize the forward KL divergence $KL(P|Q)$ between interested data distribution $P$ and parameterized density $Q$) or ELBO with empirical samples. In sampling tasks, without empirical samples, the variational approach relies on minimizing the reverse KL divergence $KL(Q|P)$. The SNF work is selected for comparison because it is originally designed for solving the sampling problem.

---

> ### Author Response · Authors · 2021-11-30
> **revision**
>
> We thank the reviewer for the feedback.
>
> I apologize that I didn’t prepare the revision. This is my first ICLR submission and I was not aware of the possibility of uploading a revision. Thanks to reviewer 1wV8 comments, I have prepared a revision today following my response that addresses most of the technical issues. Since uploading a revision is no longer is an option, I would like to share a revision through [this anonymous link](https://anonymousau.github.io/uuttqpage/assets/pdf/revision1.pdf).
>
> I sincerely hope you can take this into account in your review. We are happy to provide any further clarification and discussion.

---

### Official Review · Reviewer_d7Mk · 2021-11-08

**Correctness:** 3
**Technical Novelty And Significance:** 3
**Empirical Novelty And Significance:** 3
**Recommendation:** 6
**Confidence:** 5

**Main Review:**

The theoretical result giving the KL expression is known as acknowledged by the authors. However, to the best of my knowledge, this has never been exploited to solve sampling problems. Instead it has been proposed to instead directly approximate the Schrodinger-F\"ollmer drift. It is an interesting idea to attempt to exploit the KL expression to obtain a practical sampling algorithm.
The authors propose to parameterize the control term by some neural networks which involves the gradient of the log-target; see e.g. equation (15). The gradient of the loss is trained using recent neural SDEs techniques. The method is then demonstrated on various examples where it appears to outperform solid baselines.

I find the idea proposed in this paper interesting but I also think that a significant part of the paper was spent discussing unnecessary material and that important details were omitted. The limitations of the methodology should also be better spelled out.

First, the authors discussed a general class of controlled diffusions, review HJB for such models and Fleming logarithmic transform. It is shown that the the logarithm of the value function satisfies a simple linear PDE which can be solved using Feynman-Kac. However, these results are never used in the paper. The paper only deals with scenarios where we initialize the diffusion at the origin and consider $dX_t=u_t(X_t)dt+dW_t$ as one needs to know the marginal density of the uncontrolled process at time T. So I think it's really unnecessary to spend so much time on the general case, you might want to include this material in the supplementary and instead concentrates on more important algorithmic details. Similarly please simplify the presentation of Algorithms 1 and 2.

A key component of the paper is the parameterization $u_t(x)=NN_1(t)-NN_2(t)\nabla \log \mu(x)$ in eq. (15) of the control policy using neural networks. This should be better motivated. The use of the gradient of the log-target seems essential for the method to work well but I find it hard to believe it can solve all issues. Clearly if one has a multimodal target, gradient information will not be sufficient to explore all the modes; i.e. Langevin does not mix well in multimodal scenarios. Obviously one has also another component $NN_1(t,x)$ in the control which could deal with scenarios. However, if we are interested in sampling a high dimensional multimodal target, how can the initial control ever visit the region of high probability mass under $\mu$ and learn anything. The forward KL is the criterion been minimized so I would expect you will miss modes in some examples. In this respect, it would be good to discuss how the method is initialized (i.e. what is $u_t(x)$ at the first iteration, how is $T$ selected? etc).

The failure modes of the proposed method are never mentioned, was it necessary to train multiple times the model with various initializations to obtain good results etc, did you find examples where the method fails miserably? (this would be fine, I don't expect any method to solve all sampling problems). I think these are really important points which are unfortunately not discussed.

Similarly, I was somewhat frustrated by the lack of details on the gradient computation. Instead of just citing a paper, I would encourage the authors to discuss these details (instead of having a fairly long introduction on stochastic control). It would also be good to explain if there are any gradient issue (gradient vanishing/exploding) as the number of discretization steps increases (you seem to be using 100 in simulations, are there numerical problems when you increase the number of steps) etc.

Other questions:

- As in the AFT paper, the training time is not taking into account when evaluating the performance of the algorithm.
How large is the training time? Is it better/worse than AFT? If the training time is included, is the path integral sampler competitive to an SMC algorithm using a large number of intermediate temperatures?

- How do you select T and $\delta t$? If you initialize using $u_t(x)=0$, then at time $T$ you will have a state $X_T \sim N(0,T*I)$ but if the mode of the target is really in the tails of this distribution, would not this be a problem?

- Shouldn't the weight of interest be  $dQ^{\star}(\tau)/dQ^u(\tau)$ and not its inverse given in (17). You are sampling from $Q^u(d\tau)$ and are targetting $Q^{d\star}(\tau)$. The ESS is $\mathbb{E}_{Q^u}[{dQ^{\star}(\tau)/dQ^u(\tau)}^2]$ and not what is written in Theorem 3 given in eq. (17).

- It is claimed that a single step of the sampler in Algorithm 2 is similar to a single step of leap frog step of HMC. As the control is computed by a neural network, this really depends on the size of the network. I could not find this point discussed in the paper.

- There is quite scant information on the tuning of SMC and AFT (what flow was used? how was HMC tuned? was the Adam step size reasonable?)

- 10 temperatures as a default for SMC is certainly not standard practice. In practice, the temperature schedule is also typically selected adaptively.

- In the 2D example (Figure 1), the scale of the target is set to be larger than the initial distribution so no surprise SMC will struggle. I find the example quite misleading. The comment "The small variance is selected deliberately to distinguish the performance of the different method" in F4 should be included earlier.

- I couldn't find any details on the neural nets used in the experiment. Did you use the same architecture for all examples?

Minor comments

- SMC with annealing trick is due to Del Moral et al. (2006) not Chopin & Papaspiliopoulos (2020).

- page 13: Dai Pra (1991); ?)

- Table 1: How do you estimate the log normalizing constant for LGCP?

- Table 2: caption indicates KL-divergences but you cannot compute it as you don't have the normalizing constant.

Post-rebuttal comment: The authors didn't submit a revised version of their manuscript on time so I revised my score accordingly. I was eventually asked by the AC to consider the revised version that was sent after the deadline. As some of my comments have been taken into account, I have re-adjusted my score. In particular, I do appreciate that some of the limitations of PIS are now detailed. However, a few important points need to be addressed seriously by the authors in their final version of the paper.

- "Our approach avoid long mixing time theoretically and is more efficient": this claim is not justified and should be removed from the paper. You have no guarantee to be able to compute anything close to the optimal control and, even if you were, the Lipschitz constant of the control might be very high which would require a very fine discretization of the SDEs for numerical stability and hence a long run time. This type of bold claims is not necessary.

- Figure 1 is an unncessary ``clickbait" figure which, if kept in the manuscript, will just convince any AIS/SMC expert that you haven't run serious comparisons. The caption mentions that "The underlying distribution is chosen deliberately to distinguish performance of different methods". This is not quite correct, the problem is not the "underlying" (i.e. target) distribution but the fact that, instead of picking a diffuse initial distribution compared to the target as recommended in the SMC iterature, you pick a standard normal distribution whereas the modes of the target are at (-5,0), (-5,-5) etc. If you were using an initial distribution N(0,\sigma^2 I) with sigma=5, I am sure it'd work very well. Also the selection of 10 intermediate distributions is just arbitrary and, contrary to what the authors state, is not a recommendation made in [Arbel, 2021]. I recommend the authors to read and cite Zhou, Yan and Johansen, Adam M and Aston, John AD, Toward automatic model comparison: an adaptive sequential Monte Carlo approach, JCGS 2016. Ideally you'd run simulations with this method.

- I still believe that having a general drift and volatility in eq (2) is just unncessary when you use f=0 and g=1. Please simplify the presentation. Algorithm 1 and Algorithm 2 could be similarly simplified. It's not like you can use very general f and g in the first place as you need to know the marginal distribution of X_T for the uncontrolled SDE.

- Please cite standard references in importance sampling instead of Au & Beck, 2003. Also please do cite Neal, Annealed Importance Sampling, 2001.

- Figure 3 is another unnecessary "our method works best" figure which tells us nothing given no detail is provided (i could not find any) in the main text.

- You mentioned it in your rebuttal that "We admit a better temperature schedule may improve the performance of SMC": this is certainly true and should be stated explicitly in the paper and you should include here the reference to  Zhou, Yan and Johansen, Adam M and Aston, John AD  which addresses this problem. This is followed by a claim that you follow "the setup for SMC and AFT recommended by AFT and its official codebase". I checked the code (which is a codebase for SMC and not the official codebase) and the paper [Arbel, 2021]. You should use obviously the same settings as in this paper when you present similar examples but, if 10 temperatures is written in a code, this is not a recommendation for all examples. I think including more thorough comparisons (i.e. more temperatures, optimized schedules) would make the paper better.  You compute the "ground thruth" for the LGCP using SMC (something which should be added in the paper too and not only mentioned in the rebuttal), so it must work well sometimes.

-Section F.1 didn't address my concerns and has to be rewritten/completed.
"We note PIS training algorithm is different from AFT, where the AFT algorithm requires optimization during generating samples and
retraining the network for the next batch of samples. In contrast, the PIS control policy is trained once and used everywhere."
Once you run AFT to learn the flows, then you can rerun it with the trained flows (this is what the authors appear to be doing from my reading - "We concentrate our empirical value evaluation on the learnt flow, which is equivalent to using the test set particles." ), so it's not different from PIS.
I want to see a table with two columns, one for training where the training times for all methods are given (what is diplayed currently is the execution time for PIS and the training time for AFT) and another column where the execution times for all methods are presented  (AFT will correspond here to SMC+learned flows).





**Summary Of The Paper:**

The paper presents a control approach to sampling. It is proposed to use a controlled diffusion initialized at time t=0 to sample at a given time t=T from a given unnormalized target distribution. This is achieved by minimizing a forward KL between two suitable diffusions. The resulting expression is simple as it is given by the integrated squared control plus a final cost term. The authors propose to parameterize the control by a neural network and minimize the KL using stochastic gradient techniques. In experiments, this novel method appears to outperform some recent alternatives.

**Summary Of The Review:**

Overall, this is an interesting approach and the authors have demonstrated empirically on a variety of models that it appears to perform well.

Two points should be addressed:

- I believe that it would be good to give much more details about neural network architectures, initialization, gradient computation etc.

- The limitations of the methodology should be better spelled out. The authors should for example illustrate the failure modes of the algorithm.

More generally, it remains unclear whether any of the recent proposed Monte Carlo methods relying on training flows/neural networks by minimizing KL/maximizing ELBO are competitive with AIS/SMC when taking into account the training time.

---

> ### Author Response · Authors · 2021-11-18
> **Response for Reviewer d7Mk**
>
> We appreciate the detailed comments and feel fortunate to receive your thoughtful feedback and suggestions.
>
> **Suggestion: So I think it's really unnecessary to spend so much time on the general case, you might want to include this material in the supplementary and instead concentrate on more important algorithmic details.**
>
> Stochastic control and path integral control are key components of the derivation of the PIS algorithm and the associated theoretic results. Also, because our sampling algorithm is a stochastic control approach, we feel obligated to give a reasonable introduction to stochastic control. But we agree with the reviewer that it could be better to move some materials, including some discussions on solving PDE, derivation, and discussions of Eq 5, Eq 6 to the appendix and present main results early.
> Some practical details in Appendix G, including gradient stability and discussion of $T$, can be moved to the main paper. We will make these modifications in the revision.
>
>
> **Suggestion: A better motivation for proposed parameterization and more explanation for proposed methods, eg. more details for initialization**
>
> The advantage of PIS-Grad over PIS is clear as we show quantitative results in various tasks. We got the inspiration from the comparison between HMC and Metropolis-Hastings algorithm~[1]. The gradient information provides more efficient and promising search directions as it contains local geometrical information. An intuitive example is presented in Figure 3, where we set $u_t(x)=0$ for initialization, PIS-Grad covers more modes in limited steps while particles by PIS-NN struggle to escape the inner mode.
>
> We would like to thank the reviewer for the insightful disucssion. We tried the parametrization $u_t(x)=NN_1(t,x) + NN_2(t, \nabla \log\mu(x))$ and found it generates samples with slightly higher quality compared with the parametrization $u_t(x) = NN_1(t,x) + NN_2(t) \nabla \log\mu(x)$.
>
> From the perspective of sampling particles dynamics, existing MCMC algorithms are invariant to given target distributions. Therefore, particles are driven by gradient and random noise in a way that is independent of the given target distribution.
> PIS learns different strategies to combine gradient information and noise for different target density functions. The specialized sampling algorithm can generate samples more efficiently and shows better performance empirically in our experiments.
>
> About initialization, we use the zero initialization for the last layer of parameterized policy and other weights follow default initialization in PyTorch. Therefore, $u_t(x)=0$ for the first iteration.
>
>
> **Q: how can the initial control ever visit the region of high probability mass under $\mu$ and learn anything**
>
> There is no guarantee that PIS can cover all modes initially with only the given unnormalized distribution. However, the initialization problem exists for most sampling algorithms, eg HMC, NUTS. MCMC algorithms suffer longer mixing times compared with the ones with proper initialization. With an improper initialization and an extremely small $T$, PIS suffers from missing mode due to the proposed KL training scheme. This issue in PIS can be mitigated by increasing the value of $T$. The [Link](https://anonymousau.github.io/uuttqpage/letter/failcase/) shows uncurated samples with different $T$ and some failure cases.

---

> > ### Author Response · Authors · 2021-11-18
> > **Response for Reviewer d7Mk: algorithm details**
> >
> > **Q: The failure modes of the proposed method are never mentioned. Train multiple times with various initialization to obtain good results?**
> >
> > We will add more discussions on limitations and failure modes cases in the revision. An intuitive example has shown in the previous response. In most experiments in the paper, we found PIS with the default setting, $T=1$ and zero control initialization, gives reasonable performance. One exception is LGCP, we find training with $T=1$ sometimes suffers from numerical issues and report NAN loss. We sweep $T=1,2,5$ and found $T=5$ give encouraging results. As we discussed in Appendix G, the optimal policy $u^*$ depends on $T$ and large $T$ not only results in a large cover area but also a "smoother" $u^*$. Inspired by the reviewer's discussion, we sweep large $T$ on different target distributions with a fixed number of steps, we find training with large $T$ leads to fast convergence and covers more modes meanwhile too large $T$ can deteriorate sample quality due to large discretization error as we show in Theorem 2.
> >
> > We find PIS could fail if modes of $p(x_T)$ induced by control-free dynamics are far from away from $\mu(x)$. However, the initialization is also crucial for many MCMC approaches as we discussed above, especially the MCMC without annealing trick, including HMC and NUTS.
> >
> >
> > Among annealing-free approaches, PIS has obviously advantages with limit steps, eg $N=100$ in our experiments. Though in the paper, we present PIS as the annealing-free method, it is easy to augment PIS with the annealing trick to ease the difficulty of initialization. In PIS, we can use $(1-\alpha)\mu + \alpha N(0,1)$ as a modified target distribution and reduce $\alpha$ gradually during training. But we find without a temperating trick, PIS can already achieve competitive results.
> >
> > **Suggestion: The lack of details on the gradient computation and numerical issues**
> >
> > The gradient computation for Neural SDE can be based on stochastic adjoint sensitivity[6], which generalizes the adjoint sensitivity method for Neural ODE[5].
> >
> > We have a short discussion of Neural SDE solver in technical tips Appendix G. We use [torchsde](https://github.com/google-research/torchsde/commit/53038a3efcd77f6c9f3cfd0310700a59be5d5d2d)[2] in our implementation. Specifically, we use Euler-Maruyama without adjoint for low dimension data and the Reversible Heun method in datasets when required memory is overwhelming to cache the whole computational graph. We empirically found methods without adjoint are more stable and give lower loss compared with adjoint ones, even in low dimensional data like 2d points.
> >
> >
> > We also observed numerical instability when $T$ is very small. The magnitude and Lipchastiz constants on $t$ of optimal policy function increases when we decrease $T$. We recommend tunning $T$ when the performance is not satisfying or numerical instability occurs.
> >
> > In our experiments, we did not encounter the exploding/vanishing gradient during our experiments. As shown in [5,6], backpropagation in SDE is different from RNNs training, and backpropagation is another SDE associated with adjoint states. More discretization steps will also reduce the step size in the backpropagation SDE and have a more accurate solution for Euler-Maruyama.
> >
> > **Q: Training time**
> >
> > As discussed in Appendix F, all experiments are trained with $30$ epochs and $15000$ points datasets. The loss in most experiments plateaus after 3 epochs, sometimes even after 1 epoch. Experiments are conducted using one NVIDIA A6000 GPU. Training one epoch on 2d example takes around 15 seconds for PIS-NN and 30 seconds for PIS-Grad, 1.6 minutes and 1.8 minutes respectively on Funnel (d=10), and 7 minutes and 9 minutes on LGCP (d=1600).
> >
> > Our training algorithm is different from AFT, where the AFT algorithm requires optimization during generating samples and retraining the network for the next batch of samples. In contrast, the PIS control policy is trained once and used everywhere. Therefore, the training time can be amortized when the PIS policy is deployed in generating a large number of samples. For practical implementations with the settings reported in the paper, we show computation time for each method when generating samples with the same batch size 5000.
> >
> > | method      | Time (mean ± std) |
> > | ------ | ----------- |
> > | AFT        | 8924.2ms ± 123.17 ms        |
> > | SMC        | 130.8 ms ± 3.21 ms        |
> > | PIS-NN     | 44.8 ms ± 1.42 ms        |
> > | PIS-Grad   | 61.3 ms ± 1.51 ms        |

---

> > > ### Author Response · Authors · 2021-11-18
> > > **Response for Reviewer d7Mk: algorithm details**
> > >
> > > **Q: How do you select $T$ and $\delta t$**
> > >
> > > As we stated above, with the default $T=1$ and $u=0$ for initialization, $p(X_T)$ induced by control-free dynamics follows $N(0,I)$, which is also used for other compared approaches. An exception is LGCP as we discussed above.
> > > PIS may miss the modes if the mode of target distribution is far from the original points.
> > > We point out that the initialization is a common challenge for most methods without the annealing trick or other augmentation tricks.
> > >
> > > We use uniform discretization for $\delta t$. A potential future research direction is to cooperate the magnitude of $u_t(x)$ and the magnitude of the random noise to optimize the step size from the signal-noise ratio perspective.
> > >
> > > **Q: weights of interest**
> > >
> > > Sorry for the typo. Thanks for pointing it out. We will correct the notation and related results in the revision.
> > >
> > >
> > > **Q: As the control is computed by a neural network, this really depends on the size of the network. I could not find this point discussed in the paper.**
> > >
> > > Throughout all experiments, we use the same network with a modified first layer and last layer for different input data shape. In one pass $u_t(x)$, we augmented scalar $t$ with Fourier feature to $128$ dimension, followed by a 2 linear layers to extract $64$ dimension signal feature.
> > > We use 2 layer linear layers to extract $64$ dimension feature for $x$ and the concatenate $x,t$ features before feeding into another 3 linear layer with $64$ hidden neurons to get $u_t(x)$.
> > > A clean PyTorch implementation is included in supplementary materials under path `networks/fouriermlp.py`.
> > > We found that the training is stable with our simple MLP parametrization in our experiments and thus didn't try other neural network architectures. We will clarify this in the revision.
> > >
> > > **There is quite scant information on the tuning of SMC and AFT**
> > >
> > > The parameter of SMC and AFT are discussed in Appendix F. In experiments except for Alanine Dipeptide, we follow the setup for SMC and AFT recommended by AFT and its [official codebase](https://github.com/deepmind/annealed_flow_transport). We did little tunning for those two approaches. In the Funnel example, we find SMC and AFT with HMC kernel has reasonable performance compared with recommended slice sampling kernel and included in Table 1. Flow in AFT uses inverse autoregressive flow [3].
> > >
> > > On Alanine Dipeptide experiments, for AFT we sweep adam learning rate $lr=1e-4,5e-4,1e-3,5e-3,1e-2$ and select $5e-3$. Other settings follow the default setup from the official codebase.
> > >
> > > **Q: 10 temperatures as a default for SMC is certainly not standard practice**
> > >
> > > We admit a better temperature schedule may improve the performance of SMC. In Appendix F.4, we also conduct SMC with more intermediate temperatures and more comparison and it again shows the advantage of PIS. On the other hand, PIS can also be augmented with the annealing trick as we discussed above.
> > >
> > > **Q: misleading Figure 1, comment in F4 should be included earlier**
> > >
> > > We agree with the reviewer's comment. We will modify Figure 1 and include more details in the revision to reduce ambiguity. The curated target distribution in figure 1 is selected to show the difference and advantage of the PIS over compared algorithms in a specific setting.
> > >
> > > **Q: Did you use the same architecture for all examples**
> > >
> > > We will add a diagram for the network; we used the same network for all examples. A brief discussion is included in Appendix F.
> > > The network structure is discussed in a previous question. A clean implementation is included in the supplementary file under path `networks/fouriermlp.py`.
> > >
> > > **Q: normalization constant of LGCP**
> > >
> > > We follow the common setup in [4] as reported in Appendix F. Basically, we run SMC 100 times with 1000 intermediate temperatures to approximate the normalization constant.
> > >
> > > **Q: SMC with annealing trick citation, page 13 missing citation, KL metric in Table 2**
> > >
> > > We will correct the citation and add a mark for Table 2. The missed citation is A survey of the Schrödinger problem and some of its connections with optimal transport by Christian Leonard. We will correct it in the revision.
> > >
> > > [1]. David Mackay. Information Theory, Inference, and Learning Algorithms
> > >
> > > [2]. Patrick Kidger, James Foster, Xuechen Li, Terry Lyons. Efficient and Accurate Gradients for Neural SDEs.
> > >
> > > [3]. Diederik P. Kingma, Tim Salimans, Rafal Jozefowicz, Xi Chen, Ilya Sutskever, Max Welling. Improving Variational Inference with Inverse Autoregressive Flow
> > >
> > > [4]. Michael Arbel, Alexander G. D. G. Matthews, Arnaud Doucet. Annealed Flow Transport Monte Carlo
> > >
> > > [5]. Ricky T. Q. Chen, Yulia Rubanova, Jesse Bettencourt, David Duvenaud. Neural Ordinary Differential Equations
> > >
> > > [6]. Xuechen Li, Ting-Kam Leonard Wong, Ricky T. Q. Chen, David Duvenaud. Scalable Gradients for Stochastic Differential Equations

---

> ### Author Response · Authors · 2021-11-29
> **Revision**
>
> I apologize that I didn’t prepare the revision.
>
> This is my first ICLR submission and I was not aware of the possibility of uploading a revision. I have prepared a revision today following my response above that addresses most of the technical issues. Since uploading a revision is no longer is an option, I would like to share a revision through [this anonymous link](https://anonymousau.github.io/uuttqpage/assets/pdf/revision1.pdf).
>
> I sincerely hope you can take this into account in your review. We are happy to provide any further clarification and discussion.

---

### Official Review · Reviewer_1wV8 · 2021-11-08

**Correctness:** 3
**Technical Novelty And Significance:** 3
**Empirical Novelty And Significance:** 2
**Recommendation:** 5
**Confidence:** 3

**Main Review:**

The idea of using stochastic control for sampling is very interesting and the empirical performance seems improved. I have the below questions.

1.	Compared with previous MCMC methods, one benefit claimed in the paper is that PIS can generate samples with fewer steps. It is not clear to me why this is the case. T seems a hyperparameter in the algorithm and there are no theoretical or empirical supports to show that T is less than the number of steps in MCMC methods. Besides, what does “until converged” mean specifically in Algorithm 1? How do the authors check convergence? From Figures 1 and 2, the main advantage of PIS seems the ability of sampling from multimodal distributions. Why is it the case?

2.	In Theorem 2, the Wasserstein distance between the sampler and the target density increases as T increases which seems weird to me. Should the sampler be closer to the target distribution as the number of SDE time-discretization steps increases?

3.	Since the authors use a neural network to parameterize control policy, how does the expressiveness of neural networks affect the performance? When do the users need to include the importance sampling step? There seems no discussion about how to choose the neural networks.

4.	The empirical comparison between PIS and previous methods generally lacks interpretation. Surely the evaluation metric of PIS seems better, but it will be much helpful to provide an explanation on why it is the case. For example, is it because the target distributions are multimodal or PIS converges faster? This kind of interpretation of empirical results sheds more light on when and how to use the proposed method for users.

5.	This paper seems closely related to [Deep Generative Learning via Schrodinger Bridge, ICML 2021] which also uses Schrodinger Bridge algorithm for sampling. A discussion with this paper would be helpful.



**Summary Of The Paper:**

The authors propose a Path Integral Sampler based on Schrodinger bridge. They turn the sampling problem into a stochastic control problem and use a neural network to parameterize the control policy. In order to find the optimal control policy, several approximation techniques are introduced. The authors further propose an important sampling scheme to handle the case of a suboptimal policy. They provide convergence guarantees under Wasserstein distance and demonstrate the proposed method on synthetic datasets, alanine dipeptide and binary MNIST.

**Summary Of The Review:**

In summary I think the proposed method is interesting and technically sound. But the benefits over previous methods are not clear both in theory and practice.

---

> ### Author Response · Authors · 2021-11-18
> **Response for Reviewer 1wV8: clarifying on $T$ and more interpretation**
>
> **T seems a hyperparameter in the algorithm and there are no theoretical or empirical supports to show that T is less than the number of steps in MCMC methods.**
>
> Throughout the paper, we use $T$ to denote the time duration of the continuous-time diffusion process and $N=T/\delta t$ to denote the number of discretization steps, where $\delta t$ is the step size.
> In theory, PIS can generate perfect samples from the target distribution with optimal control policy as we show in Theorem 1 regardless of the value of $T$ in the continuous-time setting.
> However, the optimal policy depends on the choice of $T$ and small $T$ can result in large Lipschitz constant of the optimal control policy and thus numerical instability. We adopt the default $T$ value $1$ in PIS in our implementation.
> Another practical issue is the discretization error; we can only simulate discrete-time processes instead of the ideal continuous-time processes from $x_0$ to $x_T$ driven by the control policy. We bound such discretization error in Theorem 2 under condition 1.
>
> In contrast, MCMC methods use transition kernels whose invariant distribution is the target distribution and generate desired samples when $T \rightarrow \infty$. One limitation of MCMC algorithms is that they often have long mixing time and thus need a large number of iterations.
> The issue with mixing time gets worse when the target distribution is multimodal or the initialization is far from the target distribution.
>
> And based on the empirical experiments conducted in the paper, PIS generates better examples than compared with MCMC approaches with same budget of step number $N$.
> In our experiments, we compare generated samples with $N=100$ for different algorithm except for AFT and SMC which use 1000 steps. We also include experiments with large N and long run MCMC in Appendix F.4.
>
> **Besides, what does “until converged” mean specifically in Algorithm 1? How do the authors check convergence?**
>
> The training loss is defined in Eq(13). In practice, we run the training for 30 epochs and every epoch we train PIS with 15000 sampled data.
> The loss in most experiments plateaus after 3 epochs (even 1 epoch). We can basically stop training after plateau is reached, as in standard practice in neural network training. We will clarify this in the revision.
>
> **From Figures 1 and 2, the main advantage of PIS seems the ability of sampling from multimodal distributions. Why is it the case?**
>
> As discussed in the first question, one advantage of PIS is the guaranteed performance with a limited number of steps $N$. The experiments are designed to verify the strength.
>
> For multimodal distributions, MCMC has a long mixing time to reach different modes and thus needs a large number of steps. In contrast, PIS is guranteed to reach all modes in finite time if the optimal control policy is available. With the loss function in Eq(13), we empirically found that the control policy parametered by networks is close to optimal one and leads to satisfying performance.
>
> Even for one mode distribution like Funnel distribution in Sec 4.2, in our experiments PIS has a much better performance compared with MCMC methods like NUTS, HMC, SMC when the number of iterating steps is limited.
>
> From the perpective of sampling particles dynamics, existing MCMC algorithms are invariant to the target distributions. Therefore, prticles are driven by gradient of logdensity and random noise in a way that is independent of the given target distribution. By comparison, PIS learns different strategies to combine the gradient information and noise for different target densities. The specialized sampling algorithm can generate samples more efficiently and shows better performance empirically in our experiments.
>
> **In Theorem 2, the Wasserstein distance between the sampler and the target density increases as T increases which seems weird to me. Should the sampler be closer to the target distribution as  the number of SDE time-discretization steps increases?**
>
> For a fixed time horizon $T$, a larger number of SDE time-discretization step $N$ leads to a smaller $\delta t$. Since we use constant step size, the relationship can be written as $N = T / \delta t$. It is expected that a larger number of steps gives smaller error, which agrees with Theorem 2, where the loss decreases as $\delta t$ decreases.
>
> For sampling with different $T$ values, we need to take the Condition 1 into account.
> The condition assumes the optimal control policy to be Lipschitz continuous.
> For a given target distribution, the optimal policy becomes "smoother" as we increase $T$, which in turn affects the Lipchitz constants in $C_0, C_1$.
> On the other hand, for two different target distributions with different $T$ but share similar Lipchitz constants, Theorem 2 shows that the error is larger for the sampling task with a large $T$.

---

> > ### Author Response · Authors · 2021-11-18
> > **Response for Reviewer 1wV8:  more interpretation on PIS advantage**
> >
> >
> > **how does the expressiveness of neural networks affect the performance?**
> >
> > The expressiveness of neural networks determines the expressiveness of parameterized control policy.
> >
> > Across all experiments, we use the same neural network with a modified first layer and last layer for different data shapes.
> > In one forward pass for policy network $u_t(x)$, we augmented scalar $t$ with Fourier feature to $128$ dimension, followed by a 2 linear layers to extract $64$ dimension signal feature.
> > We use 2 layer linear layers to extract $64$ dimension feature for $x$ and the concatenate $x,t$ features before feeding them into another 3 linear layer with $64$ hidden neurons to get $u_t(x)$. A clean PyTorch implementation is included in supplementary materials under path `networks/fouriermlp.py`.
> > Our experiments show that the neural networks architecture we used have enough capacity and are robust for different target distributions, ranging from 2 dimension to 1600 dimension. It will be interesting to investigate the performance of other parameterizations and how network capacity affects the learned policy; it could be a possible future topic to further improve PIS.
> >
> > **When do the users need to include the importance sampling step? There seems no discussion about how to choose the neural networks.**
> >
> > When the optimal policy is not accessible, like most cases except for some simple toy examples like a mixture of Gaussian, the trained policy is sub-optimal and its generated $x_T$ is biased. To produce unbiased samples, one needs to assign the importance weights for each sample according to Eq(17).
> >
> > As we discussed in the above question, the network we used is a simple MLP with time $t$ augmented with $sin, cos$ with $64$ learned frequency. We notice that paramterizing $u_t(x)$ with $\nabla \log \mu$, which we denote as *PIS-Grad* in the paper is much better than the one without the gradient information PIS-NN.
> >
> > **The empirical comparison between PIS and previous methods generally lacks interpretation. Surely the evaluation metric of PIS seems better, but it will be much helpful to provide an explanation on why it is the case. For example, is it because the target distributions are multimodal or PIS converges faster? This kind of interpretation of empirical results sheds more light on when and how to use the proposed method for users.**
> >
> > One of biggest advantage of PIS over existing MCMC algorithms is that it does not suffer from long mixing time. The problem with long mixing time for MCMC is more serious when the target distribution is multimodal or the initiaization is not proper. The advantage is more obvious when we fix the number of time-discretization steps as we show in the experiments.
> >
> > Theorem 1 shows PIS equipped with optimal policy can reach target distribution with arbitrary choice $T$ in the continuous-time setting. In pratical discrete-time implementation, Theorem 2 bounds the discretization error. When optimal policy is not availble, Theorem 3 quanlities the sampling quality via ESS based on optimality of the learned policy.
> >
> > **This paper seems closely related to Deep Generative Learning via Schrodinger Bridge which also uses Schrodinger Bridge algorithm for sampling. A discussion with this paper would be helpful.**
> >
> > PIS shares some similarties with the suggested work. However, the task (density sampling) we investigate is different from the tasks of suggested work (density learning). In this work, PIS is learned to sample from a given unnormalized density function whose analytic formula is available, without unbiased samples from the distribution. However, density learning algorithms focus on learning unknown density with given unbiased data from the underlying distribution. In sampling tasks, due to the lack of empirical samples, PIS relies on minimizing the KL divergence $KL(Q|P)$ between interested data distribution $P$ and parameterized density $Q$ where the suggested work and many popular algorithms of density learning use negative log likelihood or $KL(P|Q)$ as objective function.

---

> ### Author Response · Authors · 2021-11-30
> **Follow up before the discussion period closes**
>
> We thank the reviewer for the feedback.
>
> I apologize that I didn’t prepare the revision. This is my first ICLR submission and I was not aware of the possibility of uploading a revision. Thanks to reviewer 1wV8 comments, I have prepared a revision today following my response above that addresses most of the technical issues. Since uploading a revision is no longer is an option, I would like to share a revision through [this anonymous link](https://anonymousau.github.io/uuttqpage/assets/pdf/revision1.pdf).
>
> I sincerely hope you can take this into account in your review. We are happy to provide any further clarification and discussion.

---

### Decision · Program_Chairs · 2022-01-20

**Decision:**

Accept (Poster)

**Comment:**

This paper introduces a control-based approach to sampling. All of the reviewers found the idea interesting. There were serious concerns by some of the reviewers regarding how the paper positioned itself relative to the literature, how it designed baselines for experiments, and how it compared itself to existing methods. There was vigorous rebuttal phase. The authors submitted a slightly late revision based on a procedural misunderstanding, and I decided to incorporate their late revision.

Based on the revision, the majority of reviewers felt that the paper was at least above the bar for acceptance and some of the more positive reviewers stood strongly by the paper. I believe that this paper is of value to the community, so I will recommend that it is accepted, but I want to be very clear about something: the authors **must** incorporate the late revision as the basis of their camera ready and I **strongly recommend** that they address the concerns of reviewer d7Mk, including but not limited to:

- "Our approach avoid long mixing time theoretically and is more efficient": this claim is too strong.
- Figure 1 is not particularly informative and the authors should reconsider it.
- The presentation of Eq (2), Algorithm 1, and Algorithm 2 should be simplified.
- Section F.1 should incorporate the comments of Reviewer d7Mk.
- Citing standard references mentioned by Reviewer d7Mk.

The reason I highlight these recommendations is that I believe they will greatly improve the quality, longevity, and impact of this paper. Slightly overselling ideas feels like a good strategy, but it is a bad long-term strategy. I believe addressing these points is in the interest of the authors.